# Context-defined cancer co-dependency mapping identifies a functional interplay between PRC2 and MLL-MEN1 complex in lymphoma

Xiao Chen [1,9,10], Yinglu Li[1,10], Fang Zhu[1,2], Xinjing Xu[1], Brian Estrella[3], Manuel A. Pazos II[3], John T. McGuire[1], Dimitris Karagiannis[1], Varun Sahu[1], Mustafo Mustafokulov[1], Claudio Scuoppo[4,5], Francisco J. Sánchez-Rivera [6,7], Yadira M. Soto-Feliciano [6,7], Laura Pasqualucci [4,5,8], Alberto Ciccia [1,4,8], Jennifer E. Amengual [3,8] & Chao Lu [1,8] ✉

Interplay between chromatin-associated complexes and modifications critically contribute to the partitioning of epigenome into stable and functionally distinct domains. Yet there is a lack of systematic identification of chromatin crosstalk mechanisms, limiting our understanding of the dynamic transition between chromatin states during development and disease. Here we perform co-dependency mapping of genes using CRISPR-Cas9-mediated fitness screens in pan-cancer cell lines to quantify gene-gene functional relationships. We identify 145 co-dependency modules and further define the molecular context underlying the essentiality of these modules by incorporating mutational, epigenome, gene expression and drug sensitivity profiles of cell lines. These analyses assign new protein complex composition and function, and predict new functional interactions, including an unexpected co-dependency between two transcriptionally counteracting chromatin complexes - polycomb repressive complex 2 (PRC2) and MLL-MEN1 complex. We show that PRC2-mediated H3K27 tri-methylation regulates the genome-wide distribution of MLL1 and MEN1. In lymphoma cells with *EZH2* gain-of-function mutations, the re-localization of MLL-MEN1 complex drives oncogenic gene expression and results in a hypersensitivity to pharmacologic inhibition of MEN1. Together, our findings provide a resource for discovery of *trans*-regulatory interactions as mechanisms of chromatin regulation and potential targets of synthetic lethality.

Chemical modifications of DNA and histones are important carriers of chromatin regulatory information that cooperate with transcription factors to integrate intrinsic and extracellular stimuli to control genome accessibility[1]. Precise regulation of chromatin dynamics is essential for maintaining cellular and organismal phenotypes and is frequently perturbed in various human diseases including cancer[2]. While modifiers (e.g. writer and eraser enzymes) and biological functions for many histone and DNA marks are well-defined individually,

less is understood about how chromatin modifications can "communicate" with each other. Enzymatic activities of chromatin modifiers are known to be sensitive to the local chromatin environment such that various *trans*-regulatory interactions exist among epigenetic marks[3–6]. These chromatin "crosstalk" are thought to contribute to the partitioning of the epigenome in a stable and sometimes heritable manner. For instance, studies have demonstrated that genome-wide patterns of DNA methylation result, at least in part, from the recruitment of de novo DNA methyltransferases guided by histone modifications[7–11]. Similarly, the establishment and propagation of histone H3K27 methylation, which is catalyzed by Polycomb Repressive Complex 2 (PRC2) and associated with transcriptional silencing, is subject to complex regulation by various chromatin-associated activities[12]. H3K36 methylation, histone marks of active transcription, allosterically inhibit PRC2's nucleosome access and catalytic activity[13,14]. The SWI/SNF chromatin remodeling complex also has an established role in antagonizing PRC2[15,16]. Accordingly, tumors harboring inactivating mutations in H3K36 methyltransferases (e.g. NSD1, SETD2) or SWI/SNF complex members exhibit globally elevated levels of H3K27 methylation[17–19]. Importantly, Tazemetostat, which inhibits the EZH2 catalytic subunit of PRC2, was recently approved to treat epithelioid sarcomas with loss of SWI/SNF complex member SMARCB1[20]. Furthermore, diffuse large B-cell lymphoma growth can be blocked by inhibition of SETD2 in the case of EZH2 hypermorphic mutation-mediated drug addiction to PRC2 inhibitors[21]. These findings suggest that disease-associated defective chromatin crosstalk could represent targets of synthetic lethality for successful therapeutic intervention.

A limited number of crosstalk between chromatin regulators and modifications have been documented through conventional biochemical approaches[7,13,22,23]. Yet for many epigenetic marks, it remains unknown if *trans*-regulatory pathways exist in addition to their direct modifiers. To date, there has been little effort to systematically survey the functional interactions among chromatin modifications and their associated complexes. Genetic interaction mapping is a powerful approach to discover gene–gene functional relationships[24]. Indeed, previous high-throughput screens have quantified the impact of pairwise gene double-knockout on the fitness of yeast strains at the genome scale to identify genetic interactions[24–26]. These efforts have successfully uncovered novel functional relationships between gene pairs, including a recent report that connects individual histone residues to chromatin-associated complexes and pathways[27]. However, a similar approach in human cells is technically challenging due to the large genome size (~20,000 genes) and thus far the largest perturbation screen only covered a small number of genetic interactions (472 genes × 472 genes)[28].

As an alternative approach, co-dependency mapping (also known as co-essentiality mapping or parallel screening) has recently been applied to identify functionally interacting genes[29–31]. Co-dependency mapping takes advantage of single gene perturbation screens in a large panel of diverse cell lines, with the assumption that two genes whose perturbations result in highly correlated phenotypes across genetically and biologically heterogeneous cell lines are likely to be functionally related. Over the past few years, several consortium-based efforts, such as DepMap[32] and Project Score[33], have performed CRISPR-Cas9 genetic knockout fitness screens in hundreds of cell lines of diverse tumor origins, providing a catalog of their genetic dependencies. Furthermore, the molecular features of these pan-cancer cell lines have been extensively profiled, generating detailed information about their genetic makeup, transcriptome, protein expression, histone modifications, and metabolome[34]. We reasoned that integrative analysis of co-dependency mapping and molecular phenotyping using these multidimensional datasets would enable systematic measurements of genetic interactions and provide a key resource to nominate new context-dependent chromatin crosstalk for mechanistic studies and translational application.

Here, we investigate the potential of combining large-scale CRISPR-Cas9 essentiality screens in >1000 pan-cancer cell lines with their molecular characteristics to define context-specific co-dependency of genes and reveal novel insights into chromatin complex composition and crosstalk. Our analysis uncovers a functional interplay between PRC2 and MLL–MEN1 complex that appears at odds with their roles in transcriptional regulation. We further investigate the impact of PRC2 and H3K27 methylation on regulating the binding of the MLL–MEN1 complex and determine the therapeutic implication of this crosstalk in the context of diffuse large B-cell lymphomas (DLBCLs) harboring gain-of-function *EZH2* mutations. Together, this work highlights the utility of integrating genetic screens with molecular profiling of cancer cells for high-throughput genetic interaction discovery that will not only offer critical insight into basic regulatory mechanisms of cancer epigenome progression but also nominate potential therapeutic targets of chromatin-associated synthetic lethality.

## Results

### Developing the genetic dependency correlation network (DCN)

Knockout of functionally linked genes is expected to produce similar fitness effects across cancer cell lines of diverse tissue origins, mutational backgrounds, and gene expression profiles. We analyzed datasets of CRISPR-Cas9 genetic perturbation screens published by Broad's Achilles and Sanger's SCORE projects from the DepMap portal[31], covering 17,386 genes across 1086 pan-cancer cell lines, to build a genetic co-dependency network. To enhance the specificity of our genetic interaction mapping, we filtered out 532 common essential genes (gene effect score < −1 in more than 90% of cell lines). For each pair of genes, we calculated the Pearson correlation coefficient for gene effect scores, which measure the effect size of gene knockout on cell fitness, across all cancer cell lines and generated the dependency correlation matrix (Fig. 1a). As a validation, we analyzed the correlation scores for genes coding for proteins with biochemical interactions documented in CORUM, a curated protein complex database[35]. We found that their scores were significantly higher than randomly selected gene pairs in equal numbers (observed vs. simulated, Supplementary Fig. 1a, b). We also overlapped the physical interactions collected in the BioGRID database with the genetic interactions in our network and 39% (305 out of 781) of our genetic interactions can be identified in BioGRID datasets (Supplementary Data 1). These results demonstrate the utility of this approach to identify complex-level protein–protein interactions.

To better visualize and prioritize significant gene–gene functional interactions, we selected genes that have at least one strong interaction with other genes (matrix cutoff, Pearson's $r > 0.4$). Distinct "modules"—groups of genes among which correlation coefficients are higher than the background—are readily discernable from the dependency correlation matrix (Supplementary Fig. 1c, Supplementary Data 2). To maximize the power of identifying functional modules, we examined the sensitivity and specificity to recover CORUM complex-level interactions using various thresholds and determined 0.34 as the optimized cutoff (Supplementary Fig. 1b). Based on this cutoff, we next generated the dependency correlation network (DCN) consisting of 145 modules (≤15 genes) (Supplementary Fig. 1d, Supplementary Data 3).

### Deplink: an integrative analysis of context-specific genetic dependency

In addition to CRISPR-Cas9 genome-wide fitness screens, cancer cell lines in the DepMap project have been extensively profiled, with detailed annotations of their genetic mutations, gene and protein expression, histone modifications, and metabolome[34]. Therefore, we sought to integrate these molecular features with our DCN analysis, in order to uncover the context underlying the essentiality of the co-dependency gene modules. We established a pipeline, named

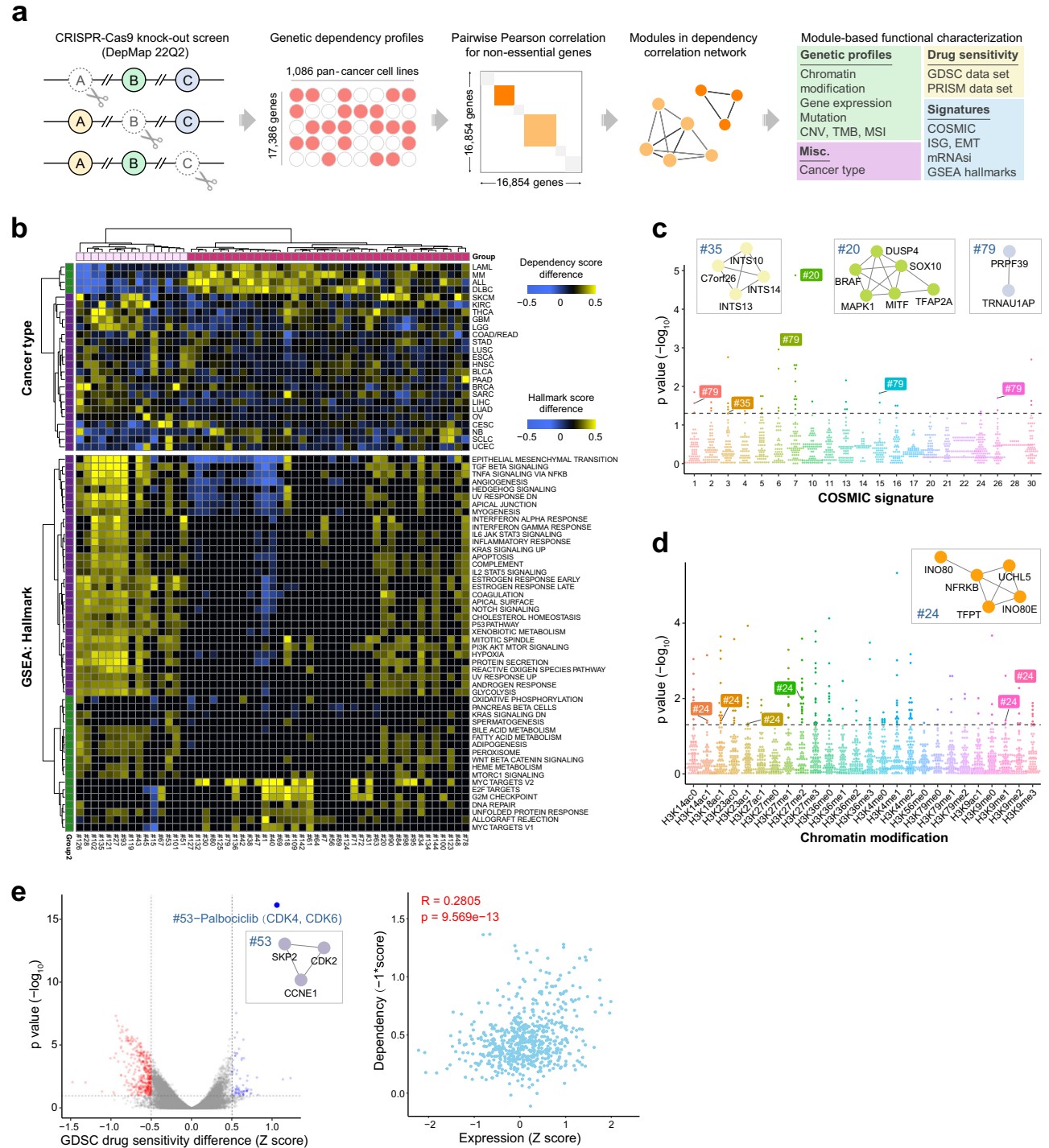

"Deplink", to identify associations between module dependencies and molecular phenotypes, including cancer type, mutational profiles, drug sensitivity, and transcriptome/chromatin signatures (Fig. 1a).

**Cancer type.** Consistent with previous report[29], we observed that some modules are selectively essential to specific cancer types (Fig. 1b). Indeed, the dependency profiles of blood cancers–acute myeloid leukemia (LAML), multiple myeloma (MM), acute lymphoblastic leukemia (ALL) and diffuse large B cell lymphomas (DLBCL)–cluster closely and away from solid tumors (Fig. 1b, Supplementary Fig. 2a). Using a cutoff of FDR < 0.1, we identified 137 cancer type-specific modules (Supplementary Fig. 2b, Supplementary Data 4). Some of these represent well-studied lineage survival oncogenes,

including *RUNX1/MYB/IRF4/NFKB1/MEF2C* (module #1) in LAML, MM and DLBC[36,37], and *MITF/SOX10* (module #20) in cutaneous melanoma[38] (Supplementary Fig. 2b); whereas others are yet to be characterized, such as *STXBP3/STX4/SNAP23* (module #15) in head and neck cancers (Supplementary Fig. 2b).

**Mutational signature.** We also identified 31 modules (*p*-value < 0.05) that are selectively essential to solid tumor cell lines carrying specific COSMIC mutational signatures[39] (Fig. 1c, Supplementary Data 5). As expected, COSMIC signature 7, predominantly found in UV exposure-linked skin cancers, predicted dependency on module #20 (*BRAF/MAPK1/MITF/SOX10*) for cell survival. These results suggest that other significantly enriched modules may be involved in the development

**Fig. 1 | Development of dependency correlation network (DCN) and Deplink analysis. a** A workflow to set up the dependency correlation network (DCN) and Deplink analysis. Step 1, acquire the dependency profiles of 17,386 genes across 1086 pan-cancer cell lines from DepMap CRISPR-Cas9 essentiality screen dataset; Step 2, for 16,854 non-essential genes, calculate the pairwise Pearson correlation score between each gene pair and generate the dependency correlation matrix; Step 3, generate DCN based on the correlation matrix using Genets; Step 4, integrate DCN with molecular profiles of pan-cancer cell lines using Deplink. **b** Heatmap showing cancer type-specific (top) or cancer hallmark gene set enrichment analysis (GSEA) signature-specific (bottom) dependency of DCN modules. Names of cancer types are consistent with TCGA study abbreviations. For cancer type-specific dependency analysis, the color scale shows the dependency score difference between cell lines from each specific cancer type and cell lines from other cancer types. 686 cell lines from cancer types containing at least 10 cell lines were analyzed. Only modules significantly associated with at least one specific cancer type are shown (FDR < 0.1). For hallmark signature enrichment analysis, the color scale

shows the hallmark signature score difference between cell lines showing high dependency for each module and cell lines showing low dependency for that module (18% top and bottom cell lines ranked by dependency score, respectively, $p$-value < 0.01). Modules are ranked in the same order for both panels. **c** Dot plot showing DCN modules that are significantly preferentially essential to solid tumor cell lines carrying various COSMIC mutational signatures. 616 cell lines from solid tumors containing at least 10 cell lines were analyzed. **d** Dot plot showing DCN modules that are significantly preferentially essential to blood cancer cell lines showing high levels of various histone modifications. 70 cell lines from blood cancers containing at least 10 cell lines were analyzed. **e** Left, volcano plot showing the correlation between the genetic dependency of DCN modules and drug sensitivity (GDSC). The cell lines with high dependency on module #53 (*CCNE1, CDK2, SKP2*) are more resistant to CDK4/6 inhibitor Palbociclib. Right, the correlation between dependency and expression of genes in module #53. For **b**–**d**, $p$ values were determined by unpaired two-tailed Student's *t*-test. For **e**, the $p$-value was determined by one-way ANOVA. Source data are provided as a Source Data file.

and/or represent vulnerabilities of specific mutational processes. For example, module #79 (*PRPF39/TRNAU1AP*) is selectively essential for cells carrying mutational signatures of defective DNA mismatch repair (COSMIC 6, 15, and 26), suggesting that this poorly characterized gene pair may play a role in the etiology or maintenance of mismatch repair deficient tumors.

**Chromatin modification.** To integrate our DCN analysis with global chromatin profiles, we identified cell lines with high abundance (top 18%) of each of the 26 histone H3 modifications quantified by Dep-Map. Seventy-four (74) modules were significantly more essential to blood cancer cell lines carrying specific chromatin signatures (Fig. 1d, Supplementary Data 6). Notably, H3K14ac-, H3K18ac- and H3K23ac-high cell lines share a common dependency on module #24, which consists of genes coding for INO80 chromatin remodeling complex (Fig. 1d). To probe the potential underlying mechanism, we analyzed genome-wide binding of INO80 complex in relation to various chromatin features using publicly available datasets[40]. INO80 complex localization positively correlated with open chromatin, histone acetylation, and the binding of various histone acetyltransferases (Supplementary Fig. 2c). Therefore, it is possible that cells with histone hyperacetylation recruit INO80 complex and require its nucleosome-remodeling activity to maintain genome access and/or integrity.

**Transcriptome signature.** By integrating the genetic co-dependency maps and gene expression profiles, we annotated modules that were preferentially essential to cell lines exhibiting various cancer hallmark transcriptional signatures (Fig. 1b, Supplementary Data 7). Several significant correlations emerged from this analysis that are consistent with the described functions of module genes. For example, module #51 (*DCP2/XRN1/ADAR*) and module #101 (*FOXA1/SPDEF*) are more essential to cells displaying transcriptional signatures of "Interferon alpha response" and "Estrogen Response", respectively[41–43]. We also found that functionally linked gene expression signatures, such as "Epithelial–Mesenchymal Transition" and "TGF beta signaling", were closely clustered based on module effect scores (Fig. 1b). Therefore, gene expression signatures represent another important determinant for the specificity of module dependencies.

**Drug response.** We incorporated data from Genomics of Drug Sensitivity in Cancer (GDSC)[44] to determine the relationship between DCN module dependencies and drug response. For each DCN module, we quantified the differences in drug sensitivity ($IC_{50}$) between dependent vs. non-dependent cell lines. Using a cutoff of drug sensitivity *Z*-score > 0.2 and FDR < 0.1, we uncovered 1262 significant drug–module pairs (Supplementary Data 8). Importantly, cells that

required module #20 (*BRAF/MAPK1*) for survival also displayed increased sensitivity to multiple inhibitors of the BRAF/MEK pathway, providing validation for our analysis (Supplementary Fig. 2d). Intriguingly, we found that cells dependent on module #53 (*CCNE1/SKP2/CDK2*) are highly resistant to CDK4/6 inhibitor Palbociclib (Fig. 1e). This finding is consistent with previous studies reporting the link between *CCNE1* (cyclin E) overexpression and Palbociclib resistance[45,46]. Indeed, we confirmed that there was a significant positive correlation between *CCNE1/SKP2* expression and dependency (Fig. 1e). These results suggest a potential synergy between inhibitors of cyclin E and CDK4/6 in cancer treatment.

## DCN reveals novel insights into composition of chromatin complexes

We next examined chromatin-related DCN modules in further detail. Consistent with our global analysis (Supplementary Fig. 1), many modules correspond to chromatin biochemical complexes documented in the CORUM database, including EMSY complex (module #21), SIN3–HDAC complex (module #73), G9a/GLP H3K9 methyltransferase complex (module #49), ANCO1–HDAC3 complex (module #62), INO80 complex (module #24) and Integrator complex (module #35) (Fig. 2a). Interestingly, SIN3A and SIN3B were identified in distinct DCN modules (SIN3–HDAC and EMSY, respectively), highlighting a functional divergence between the two SIN3 family members[47].

Some modules contain genes that are not annotated as members of the corresponding chromatin complexes, such as *SETD5* in module #62/ANCO1–HDAC3 complex[48–50] and *C7ORF26* in module #35/Integrator complex (Fig. 2a). In particular, *C7orf26* shows similar dependency landscape to *INTS10/13/14*, but not genes encoding other integrator complex subunits, across 1086 pan-cancer cell lines (Supplementary Fig. 3a). We tested if these genetic co-dependencies represent biochemical interactions. We ectopically expressed FLAG-tagged C7orf26 in HeLa cells and performed co-immunoprecipitation followed by mass spectrometry. A specific interaction between C7orf26 and INTS10/13/14, but not other Integrator complex members, was identified and validated using immunoblotting (Supplementary Fig. 3b–d, Supplementary Data 9). These data indicate that C7orf26 and INTS10/13/14 may form a biochemically distinct subcomplex that is functionally independent of the canonical Integrator complex. In agreement, Deplink analysis revealed that C7orf26-containing module #35 was preferentially essential to solid tumor cell lines exhibiting COSMIC mutational signature 3 (failure of DNA homologous recombination repair) (Fig. 1c and Supplementary Fig. 3e). Therefore, C7orf26/INTS10/13/14 subcomplex may participate in the DNA damage response independently of the canonical function of Integrator complex in transcriptional regulation.

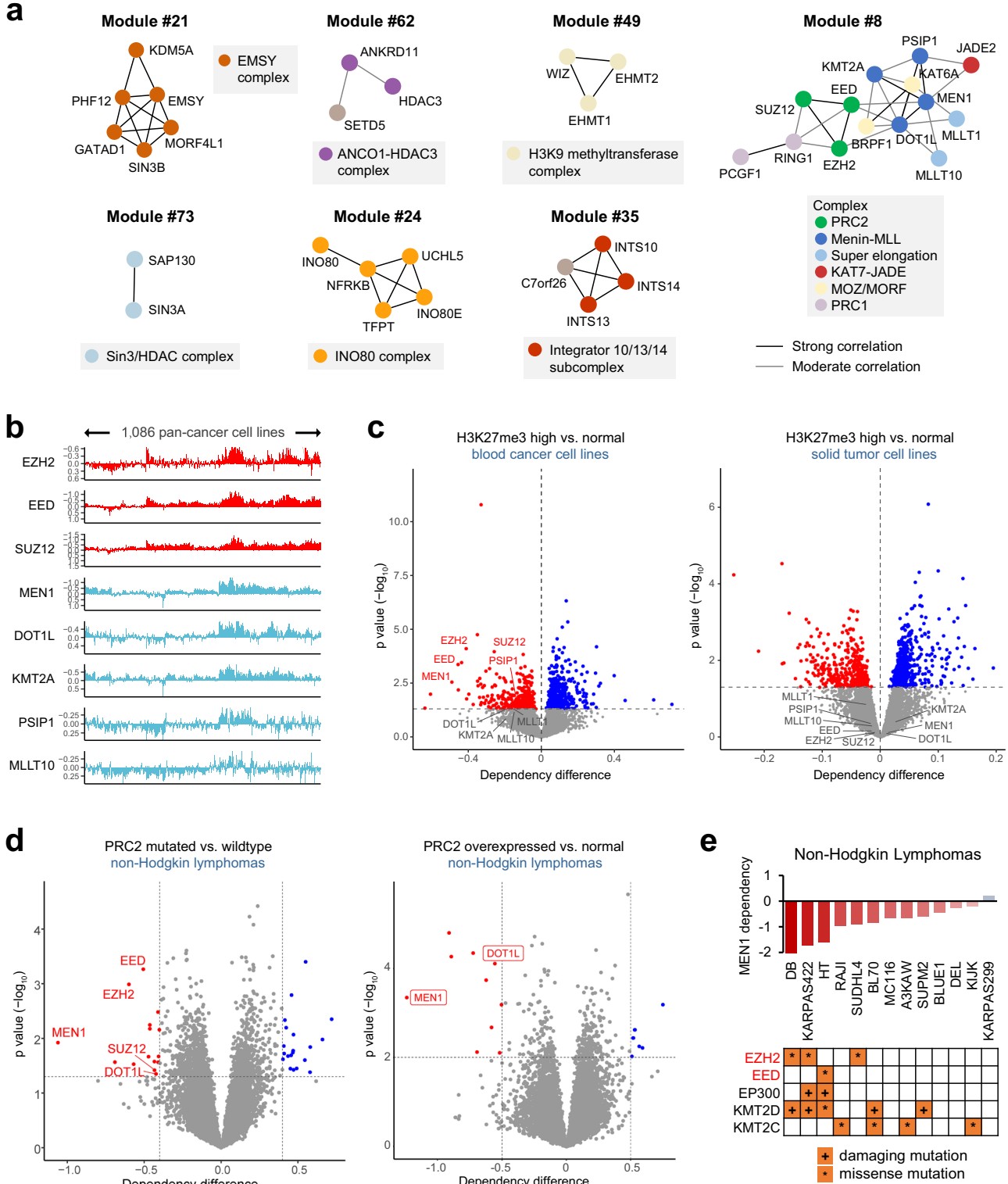

**Fig. 2 | Deplink reveals novel insights into the composition and interaction of chromatin complexes. a** Representative chromatin-associated DCN module. The darkness of lines between genes indicates the correlation strength. Known members from the same complex are denoted as dots in same color. **b** Dependency landscape of PRC2 and MLL–MEN1 complex members across pan-cancer cell lines (*x*-axis). 1086 pan-cancer cell lines were analyzed. *Y*-axis indicates gene effect scores. Low score indicates that a gene is more likely to be essential in a given cell line. **c** Left, volcano plot showing the difference in genetic dependency between blood cancer cell lines (*n* = 70 cell lines) with high vs. normal levels of H3K27me3. Right, volcano plot showing the difference in genetic dependency between solid

tumor cell lines (*n* = 616 cell lines) with high vs. normal levels of H3K27me3. **d** Left, volcano plot showing the difference in genetic dependency between non-Hodgkin lymphoma cell lines carrying PRC2 gene mutations (*n* = 5 cell lines) and PRC2-wildtype non-Hodgkin lymphoma cell lines (*n* = 9 cell lines). Right, volcano plot showing the difference in genetic dependency between non-Hodgkin lymphoma cell lines with high (*n* = 3 cell lines) vs. normal level (*n* = 11 cell lines) of PRC2 gene expression. **e** Bar plot showing the *MEN1* dependency in 13 non-Hodgkin lymphoma cell lines. For each cell line, mutations in chromatin enzymes are indicated in the table below. For all experiments, *p* values were determined by unpaired two-tailed Student's *t*-test. Source data are provided as a Source Data file.

## Functional interplay between MLL−MEN1 complex and PRC2 complexes

We speculate that our DCN analysis may also reveal functional relationships between biochemically distinct chromatin complexes. To this end, we focused on module #8, which contains MLL(KMT2A)-MEN1, PRC1, PRC2, KAT7-JADE and MOZ/MORF (KAT6A/B) complexes (Fig. 2a, b). This co-dependency module is also intriguing since it contains well-characterized chromatin complexes with distinct and opposing functions in either transcriptional activation (MLL−MEN1, KAT7, and KAT6A/B) or repression (PRC1, PRC2). Deplink analysis found that genes within module #8 were preferentially essential to blood but not solid cancer cell lines with high levels of H3K27me3 (Fig. 2c, Supplementary Fig. 4a). While it is conceivable that H3K27me3-high cell lines, such as DLBCL cells carrying EZH2 gain-of-function mutations[51], require PRC2 complex for survival[52], the role of MLL−MEN1 complex in this setting is unclear. We confirmed that DLBCL cell lines with either mutations or overexpression of PRC2 complex members were significantly more dependent on MEN1 and DOT1L for survival (Fig. 2d, e).

## EZH2 mutant DLBCL cells display preferential dependency on MLL-MEN1

To validate findings from our DCN and Deplink analysis, we independently performed a CRISPR genetic screen using a chromatin-focused gRNA library[53] in a Cas9-expressing EZH2 mutant DLBCL cell line (KARPAS422) (Fig. 3a). We found highly concordant effects of knocking out PRC2 complex members (EZH2/EED/SUZ12), MLL1 and MEN1 on decreasing cell competitive fitness between our screen and the DepMap dataset (Fig. 3b, c). As a comparison, knockout of Module #8 genes had minimal impact on cell fitness in K562 leukemia cell line that is EZH2 wildtype (Supplementary Fig. 4b). The detrimental effect of CRISPR-Cas9-mediated knockout of MEN1 on the proliferation of EZH2 mutant lymphoma lines KARPAS422 and SuDHL4 was also evident in competitive proliferation assays (Fig. 3d), and in xenograft experiments where depletion of MEN1 decreased the incidence of tumor formation (Fig. 3e). Furthermore, we subjected a panel of lymphoma cell lines to a small-molecule inhibitor of MEN1, MI-503[54], and observed that compared to EZH2 wildtype, EZH2 mutant cells were significantly more sensitive to MEN1 inhibition (Fig. 3f, Supplementary Fig. 4c).

In addition to DLBCL, a subset of multiple myeloma (MM) tumors display globally elevated H3K27me3 levels due to either inactivation of H3K27 demethylases (UTX/KDM6A) or overexpression of PHF19[55,56]. H3K27me3-high MM cell lines also show increased sensitivity to inhibition of EZH2[55–57]. We treated a pair of MM cell lines that are either H3K27me3-high (RPMI-8226) or H3K27me3-low (MM.1S) (Supplementary Fig. 4d) with MEN1 inhibitor (VTP50469)[58]. MEN1 inhibition significantly inhibited the viability of RPMI-8226 but not MM.1S cells in a dose-dependent manner (Fig. 3g). This finding further confirms our Deplink analysis result that the co-dependency between PRC2 and MLL−MEN1 applies broadly to hematopoietic tumors with high levels of H3K27me3 (Fig. 2c).

## Expansion of H3K27me3 domains drives redistribution of MLL−MEN1 complex

To investigate how PRC2 and MLL−MEN1 complexes, which are involved in transcriptional repression and activation, respectively, demonstrate cooperative effects on lymphoma cell fitness, we performed CUT&Tag to profile genome-wide distribution of H3K27me3, MEN1, and MLL1 in both EZH2 wildtype (Farage) and mutant (KARPAS422) lymphoma lines. As expected, genome-wide bindings of MEN1 and MLL1 are strongly correlated and enriched at gene promoter regions (Supplementary Fig. 5a, b, Supplementary Data 10). Consistent with previous reports[59], there was an increase in the number of large (>100 kb) H3K27me3 domains in EZH2 mutant KARPAS422 cells compared to the wildtype Farage cells (Fig. 4a), suggesting that EZH2 gain-of-function mutation

promotes the expansion of H3K27me3 domains. In contrast, the number of total or promoter-specific MEN1 and MLL1 peaks was markedly reduced in KARPAS422 cells compared to Farage cells (Fig. 4b, Supplementary Fig. 5c, d). By integrating these datasets, we found that the changes in H3K27me3 were significantly negatively correlated with changes in MLL1/MEN1 bindings (Fig. 4c, Supplementary Fig. 5e). Furthermore, these changes were associated with changes in gene expression, with promoters that gained H3K27me3 and lost MLL1/MEN1 bindings exhibited decreased expression of corresponding genes (Fig. 4c, Supplementary Fig. 5e). As representative examples, H3K27 was hypermethylated at genomic loci encompassing MYB and IRF5 genes in KARPAS422 cells, accompanied by abolished MEN1/MLL1 binding and silencing of these two genes (Fig. 4d). At the CD24 loci, on the other hand, loss of H3K27me3 from the promoter region was correlated with MEN1/MLL1 binding and CD24 transcription. These results prompted us to consider the possibility that the expansion of H3K27me3 by hyperactive mutant EZH2 prohibits MLL1/MEN1 binding, resulting in a genome-wide redistribution and "concentration" of MLL−MEN1 complex at a limited number of accessible gene promoters. Supporting this notion, MLL1/MEN1 promoter peaks present in both KARPAS422 and Farage cells showed markedly higher signal abundance (Fig. 4e) and higher expression of their associated genes (Fig. 4f) in KARPAS422 cells.

To determine if similar H3K27me3-driven redistribution of MLL−MEN1 complex binding can be observed in MM cells, we performed epigenomic profiling of H3K27me3-high RPMI-8226 and H3K27me3-low MM.1S cells. We found an increased number of H3K27me3 large domains in RPMI-8226 cells, which was accompanied by a decrease in the number of MEN1 but not MLL1 peaks (Supplementary Fig. 6a, b). Importantly, both genome-wide analysis and inspection at representative loci revealed higher signal abundance for MEN1/MLL1 promoter peaks in RPMI-8226 cells (Supplementary Fig. 6c, d), mirroring that in KARPAS422 cells.

We next determined if higher levels of MLL1/MEN1 binding in H3K27me3-high DLBCL or MM cells can be reversed with depletion of H3K27me3. To this end, we treated KARPAS422 cells with an EZH2 inhibitor (EPZ-6438). EZH2 inhibition effectively abolished genome-wide H3K27me3 enrichment, and regained MLL1/MEN1 bindings at H3K27me3-high regions in untreated cells (Supplementary Fig. 7a, b). Concomitantly, we observed decreased signal abundance of shared MLL1/MEN1 peaks (Fig. 4e, Supplementary Fig. 7a, b). A similar "titration" of MLL1/MEN1 binding was also observed in RPMI-8226 cells following EPZ-6438 treatment (Supplementary Fig. 6c). Together, these results suggest a causal function of H3K27me3 in regulating genome-wide patterns of MLL1/MEN1 binding.

## EZH2 mutant lymphoma cells are addicted to MLL−MEN1 target genes

We reason that expression of genes bound by high levels of MEN1/MLL1 is also hypersensitive to MEN1 inhibition, which could underline the preferential toxicity of MEN1 inhibitor to EZH2 mutant lymphoma cell lines. We assessed MEN1/MLL1 and H3K27me3 enrichment and gene expression in Farage and KARPAS422 cells before and after the treatment of MEN1 inhibitors MI-503 and VTP50469 using CUT&Tag and RNA-seq, respectively. We observed a gain of H3K27me3 surrounding MLL1/MEN1 binding peaks following MEN1 inhibitor treatment in both KARPAS422 and Farage cells (Supplementary Fig. 7c–e), suggesting that the opposition between MLL-MEN1 complex and H3K27me3 is bidirectional. MEN1 inhibitor treatment resulted in a more pronounced loss of MEN1 and MLL1 bindings to gene promoters in KARPAS422 cells compared to Farage cells (Fig. 5a, Supplementary Fig. 8a, b). This preferential loss of MEN1/MLL1 binding was associated with significantly reduced transcription of MLL1/MEN1-bound genes and a larger impact on the transcriptome of KARPAS422 cells following MI-503 exposure (Fig. 5a, Supplementary Fig. 8c, Supplementary Data 11). We identified 60 genes that were bound by MEN1 and

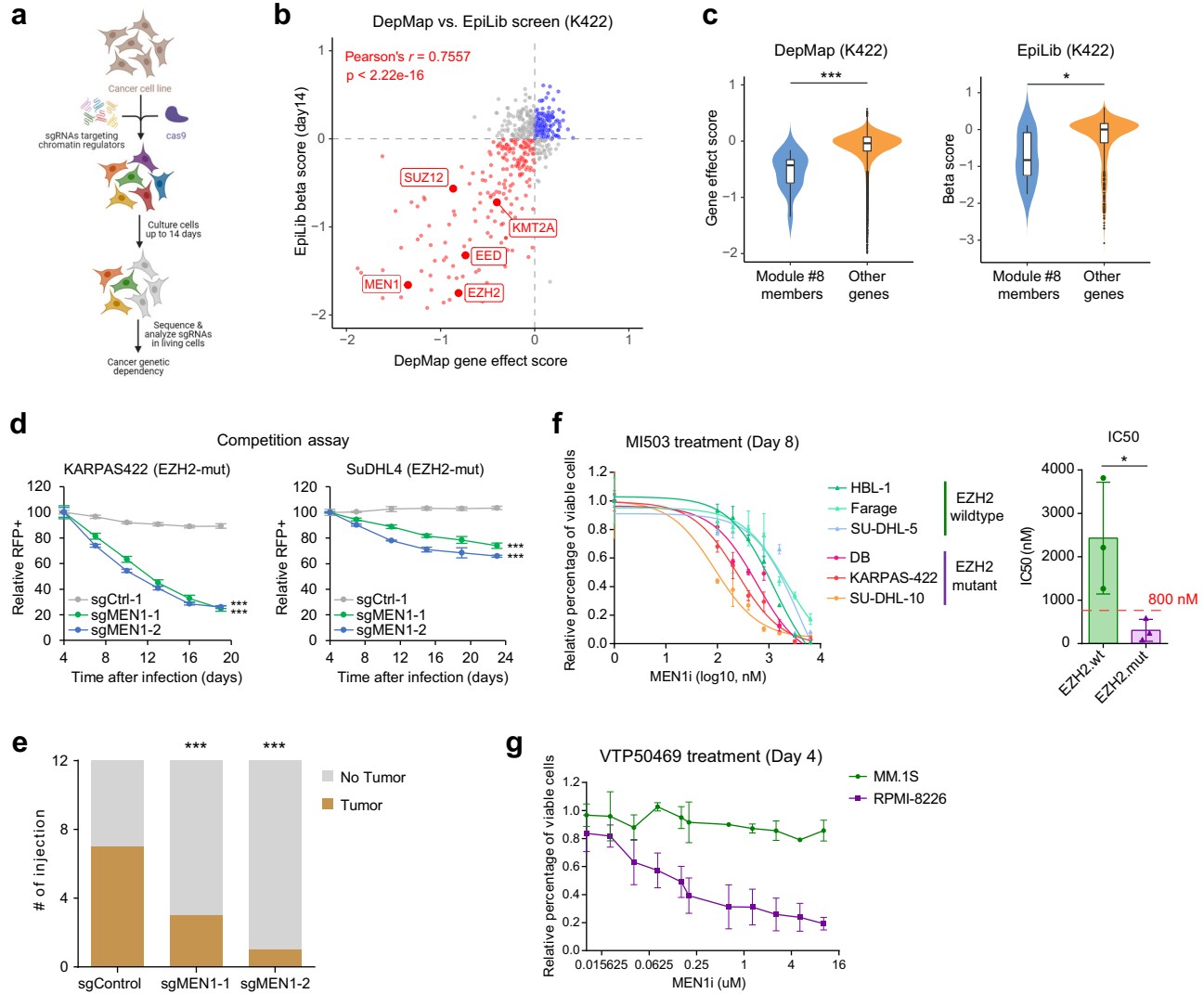

**Fig. 3 | CRISPR-Cas9 essentiality screens reveal preferential dependency of MLL–MEN1 in *EZH2*-mutated DLBCL. a** A schematic diagram showing the workflow of chromatin-focused CRISPR-Cas9 essentiality screen (EpiLib screen) in DLBCL cell lines. Created by BioRender. **b** Scatter plot showing *MLL1*, *MEN1*, and PRC2 complex members are highly essential to DLBCL cell line KARPAS422 (K422) in both EpiLib screen and the genome-wide CRISPR-Cas9 essentiality screen from DepMap. *p*-value was determined by one-way ANOVA. **c** Violin plots showing the significant differences in dependency between genes in module #8 (*n* = 14 and 12 genes for DepMap and EpiLib screens, respectively) and other genes (*n* = 16,840 and 553 genes for DepMap and EpiLib screens, respectively) in DepMap screen (left) and EpiLib screen (right) in *EZH2* mutant KARPAS422 (K422) cells. Lower gene effect score indicates that a gene is more likely to be essential. The center line in the embedded boxplots represents the median, the box limits are the 25th and 75th percentiles, and the whiskers are the minimum to maximum values. *P* values were determined by unpaired two-tailed Student's *t*-test. For statistical analyses of DepMap and EpiLib screens, *p* = 2.38e−4 and 0.0267, respectively. **d** Negative-selection competition assay measuring the relative percentage of RFP⁺/sgRNA-positive cells over time after transduction of *EZH2* mutant DLBCL cell lines

KARPAS422 and SuDHL4 with control or *MEN1* sgRNAs. *n* = 3 independent experiments. Dots and whiskers are mean ± s.d. *P* values were determined by paired two-tailed Student's *t*-test. For statistical analyses of KARPAS422 sgMEN1-1/−2, *p* = 8.24e-6 and 1.54e-6, respectively. For statistical analyses of SuDHL4 sgMEN1-1/−2, *p* = 1.11e−5 and 3.18e-6, respectively. **e** Incidence of xenograft tumor formation after injection of KARPAS422 cells transduced with control or *MEN1* sgRNAs. *P* values were determined by Fisher's exact test. For statistical analyses of sgMEN1-1/−2, *p* = 3.53e−6 and 1.54e-14, respectively. The experiments contain 12 injections for each condition and are not repeated. **f** Relative percentage of viable cells after 8-day treatment of MEN1 inhibitor MI503 with indicated dosage (left) and measured IC50 (right) for *EZH2*-mutant and -wildtype DLBCL cell lines. *n* = 3 independent experiments. Dots and whiskers are mean ± s.d. *P* values were determined by unpaired two-tailed Student's *t*-test. For statistical analyses of IC50, *p* = 0.0488. **g** Relative percentage of viable cells after 4-day treatment of MEN1 inhibitor VTP50469 with indicated dosage for H3K27me3-high (RPMI-8226) and H3K27me3-low (MM.1S) multiple myeloma cell lines. *n* = 3 independent experiments. Dots and whiskers are mean ± s.d. **p* < 0.05; ***p* < 0.01; ****p* < 0.001. Source data are provided as a Source Data file.

significantly downregulated after MI-503 treatment in KARPAS422 cells. Approximately one-third (21) of these genes were also downregulated by MI-503 in another *EZH2* mutant lymphoma cell line DB. Importantly, genes that were downregulated following MEN1 inhibition showed significantly lower dependency scores in genome-wide knockout screens of KARPAS422 and DB cells, suggesting that MEN1-regulated genes are more essential to the fitness of *EZH2* mutant DLBCL cell lines (Fig. 5b).

Gene Ontology (GO) analysis of MEN1-regulated genes revealed a significant enrichment of mTORC1 signaling and unfolded protein response (UPR) pathways (Fig. 5c). Indeed, levels of phosphorylated S6, a marker of mTORC1 signaling, were higher in KARPAS422 cells and suppressed by MI-503 treatment (Fig. 5d). Closer examination of individual genes confirmed results of genome-wide analysis: compared to Farage cells, these genes exhibited decreased occupancy of H3K27me3, increased binding of MEN1 at their promoters, and

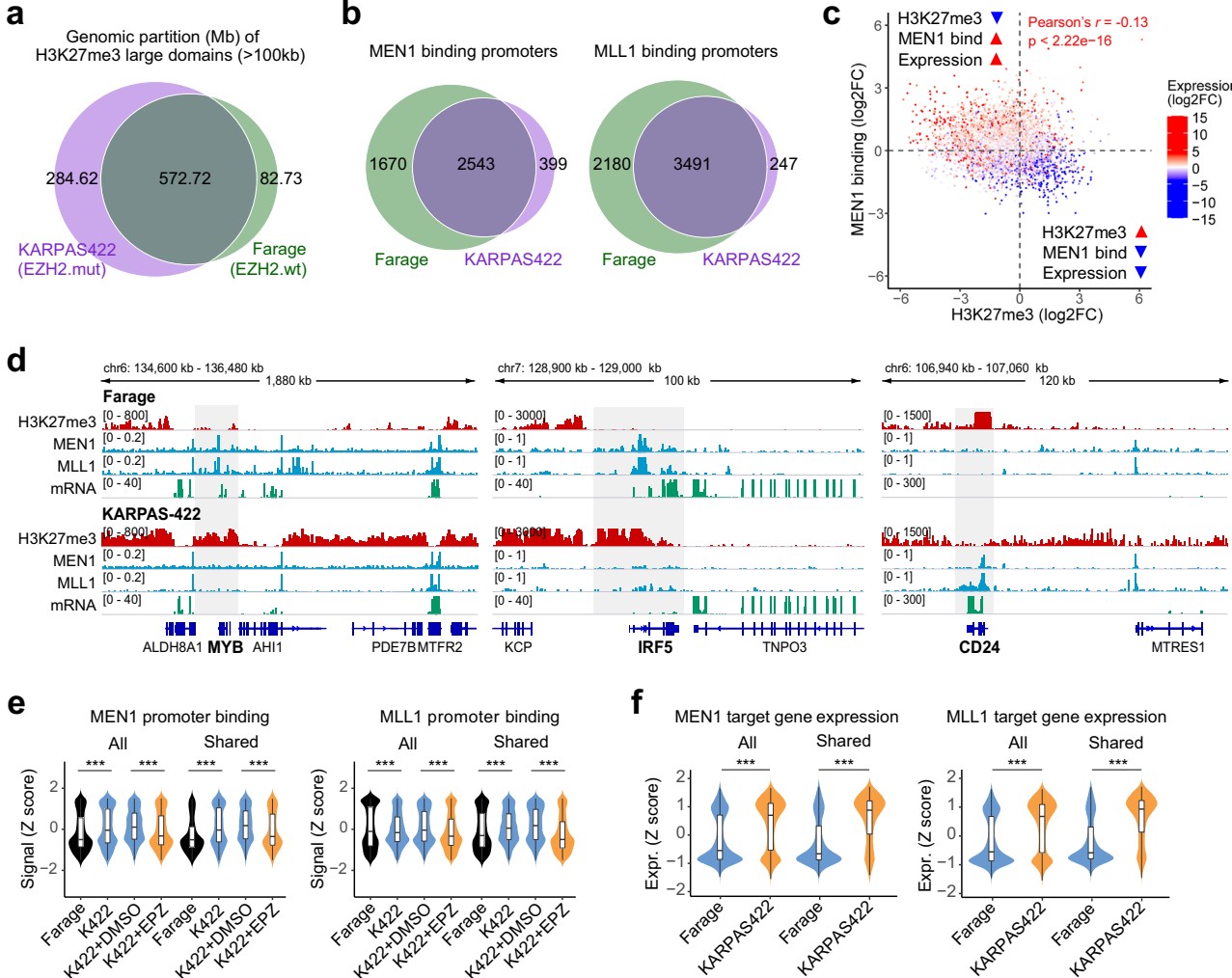

**Fig. 4 | Expansion of H3K27me3 domains in *EZH2*-mutant cells drives redistribution of MLL–MEN1 complex. a** Venn diagram showing the genomic partition (Mb) of H3K27me3 large domains (>100 kb) shared between *EZH2* mutant DLBCL cell line KARPAS422 (purple) and *EZH2* wildtype cell line Farage (green). **b** Venn diagrams showing the MEN1 (left) or MLL1 (right) binding promoters shared between KARPAS422 (purple) and Farage (green) cells. **c** Density plot showing the correlation between the differences in H3K27me3 abundance (*x*-axis), MEN1 binding (*y*-axis) at gene promoter regions, and corresponding gene expression changes (log2 of fold change, colored in blue and red for down- and upregulation, respectively) between KARPAS422 and Farage cells. **d** Integrative Genomics Viewer snapshot showing the landscape of H3K27me3, MEN1, and MLL1 binding, and gene transcription at the *MYB* (left), *IRF5* (middle), and *CD24* (right) gene loci in DLBCL cell lines. **e** Violin plots showing the normalized signal abundance (*Z* score) of MEN1 (left) and MLL1 (right) peaks at all MEN1/MLL1-bound promoters in Farage and KARPAS422 cells (all, *n* = 5075 and 6813 for all MEN1/MLL1-bound promoters, respectively), or only those MEN1/MLL1-bound promoters shared between Farage and KARPAS422 cells (shared, *n* = 2305 and 3562 for shared MEN1/MLL1-bound

promoters, respectively). EPZ, EZH2 inhibitor treatment using 1 μM EPZ-6438 for 72 h. The center line in the embedded boxplots represents the median, the box limits are the 25th and 75th percentiles, and the whiskers are the minimum to maximum values. For statistical analyses of MEN1 promoter binding, *p* < 2.22e−16, *p* < 2.22e−16, *p* < 2.22e−16, and *p* = 2.96e−11, respectively. For statistical analyses of MLL1 promoter binding, *p* = 1.44e−4, *p* < 2.22e−16, *p* < 2.22e−16, and *p* < 2.22e−16, respectively. **f** Violin plots showing the normalized gene expression (*Z* score) of all MEN1 (left) and MLL1 (right) target genes (all, *n* = 5075 and 6813 for all MEN1 and MLL1 target genes, respectively), or only those MEN1/MLL1 target genes shared between Farage and KARPAS422 cells (shared, *n* = 2305 and 3562 for shared MEN1 and MLL1 target genes, respectively). The center line in the embedded boxplots represents the median, the box limits are the 25th and 75th percentiles, and the whiskers are the minimum to maximum values. For statistical analyses of both MEN1/MLL1 target gene expression, *p* < 2.22e−16 in all comparisons. For all experiments, *p* values were determined by unpaired two-tailed Student's *t*-test. ****p* < 0.001. Source data are provided as a Source Data file.

increased transcription in KARPAS422 cells (Fig. 5e). Upon MEN1 inhibition, however, there was a more pronounced decrease in the MLL1/MEN1 binding at, and the expression of, these mTORC1 signaling pathway genes in *EZH2* mutant KARPAS422 and DB cells (Fig. 5f, Supplementary Fig. 8d). Notably, two genes showing the largest decrease in expression upon MEN1 inhibition, *PSAT1* and *SLC1A5*, were significantly associated with worse prognosis in a DLBCL patient cohort[60] (Supplementary Fig. 8e). Furthermore, compared to Farage cells, KARPAS422 cells were more sensitive to pharmacologic inhibition of mTORC1 by rapamycin (Fig. 5g).

Conversely, knockout of *TSC1*, an upstream negative regulator of mTORC1 signaling, partially rescued the decreased viability of KARPAS422 cells following VTP50469 treatment (Supplementary Fig. 8f), supporting a functional role of mTORC1 signaling in *EZH2* mutant DLBCL cells' hypersensitivity to MEN1 inhibition. Taken together, it appears that the redistribution of the MLL–MEN1 complex by H3K27me3 expansion drives a higher and MEN1-dependent expression of genes involved in key oncogenic pathways (e.g. mTORC1 signaling) that are required for the growth of *EZH2* mutant DLBCL cells (Fig. 5h).

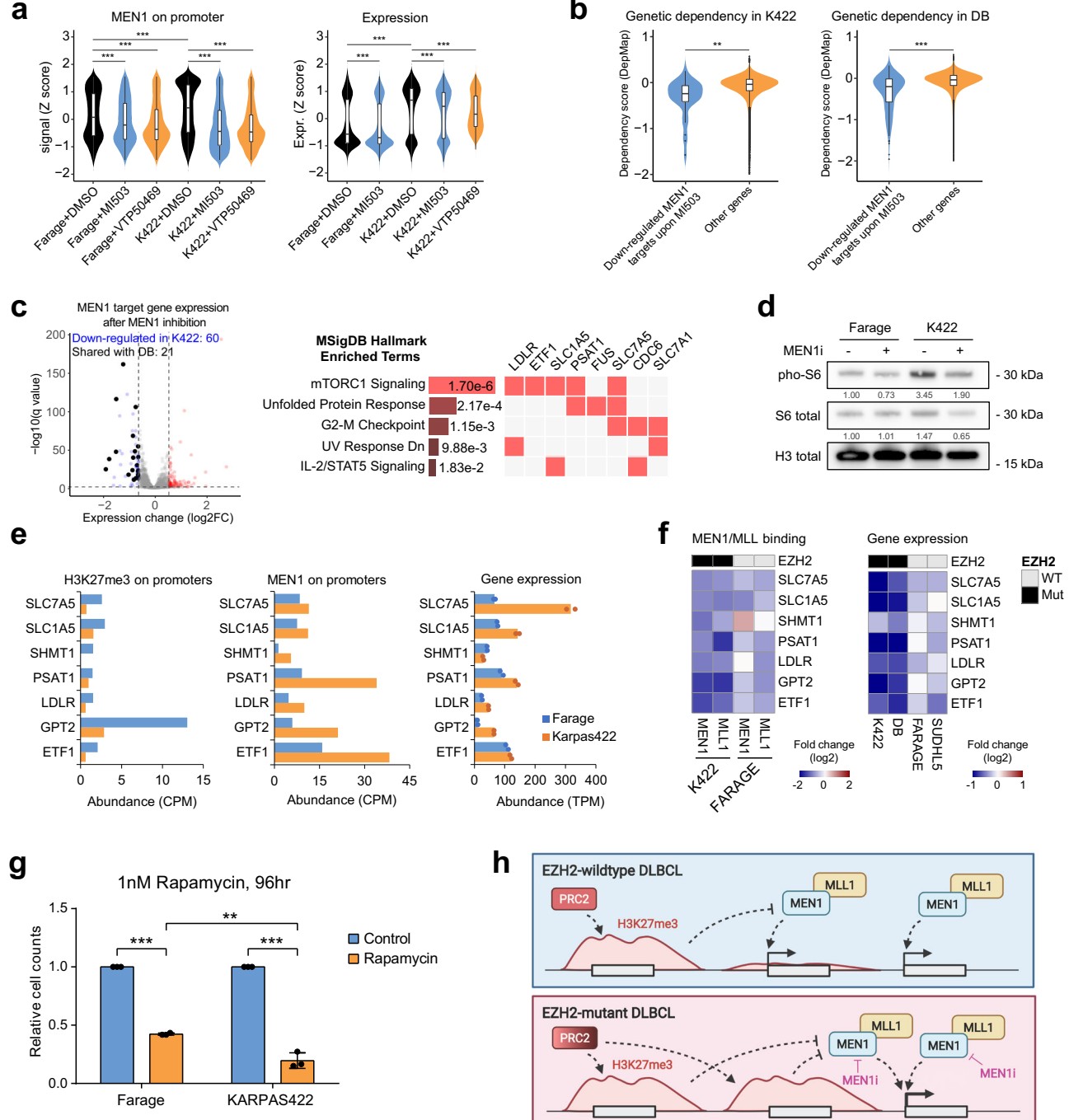

## MEN1 inhibitor synergizes with Tazemetostat in inhibiting growth of *EZH2* mutant DLBCL

Tazemetostat (EPZ-6438) is an EZH2 inhibitor recently approved for relapsed or refractory follicular lymphomas carrying *EZH2* gain-of-function mutations[61]. However, the clinical benefit of Tazemetostat as a single agent for *EZH2* mutated DLBCL is modest[62]. We, therefore, tested if combined inhibition of MEN1 (MI-503 or VTP50469) and EZH2 (EPZ-6438) could result in synergistic effects on tumor transcriptome and growth in vitro and in vivo. RNA-seq analysis indicated that while VTP50469 and EPZ-6438 each had modest impact on gene expression of KARPAS422 cells, co-treatment of both drugs induced >2400 differentially expressed genes (914 upregulated and 1513 downregulated) which were functionally enriched for Myc targets, E2F targets as well as mTORC1 signaling (Fig. 6a, Supplementary Fig. 9a). This synergy on transcriptome remodeling was also evident in another *EZH2* mutant

DLBCL line SuDHL10, where VTP50469 had minimal impact as single treatment yet together with EPZ-6438 caused differential expression of >600 genes (Fig. 6b, Supplementary Fig. 9b). Consistently, a synergistic impact on decreasing cell viability between EZH2 and MEN1 inhibitors was observed in KARPAS422 and SuDHL10 but not Farage cells (Fig. 6c, Supplementary Fig. 9c). A similar trend was found in MM, where co-treatment of EPZ-6438 + VTP50469 decreased the viability of H3K27me3-high RPMI-8226 cells but not H3K27me3-low MM.1S cells (Supplementary Fig. 9d). We also tested the efficacy of combination therapy in DLBCL cells that have acquired resistance to EZH2 inhibition. We continuously exposed the *EZH2*-mutant lymphoma cell line DB to increasing concentrations of EPZ-6438 for 6 weeks to develop an EPZ-6438-resistant line (DB-r) (Supplementary Fig. 9e). Combined treatment of MI-503 and EPZ-6438 displayed a strong synergistic effect on inhibiting the proliferation of DB-r cells (Fig. 6d). Finally, to test the

**Fig. 5 | *EZH2* mutant lymphomas are addicted to MLL–MEN1-regulated oncogenic gene expression. a** Violin plots showing the normalized signal abundance (*Z* score) of MEN1 peaks at MEN1-bound promoters (left, *n* = 5075 promoters) and normalized gene expression (*Z* score) of MEN1 target genes (right, *n* = 5075 genes) in Farage and KARPAS422 cells treated with DMSO or MEN1 inhibitors (800 nM MI-503 for 3 days, 330 nM VTP50469 for 7 days). The center line in the embedded boxplots represents the median, the box limits are the 25th and 75th percentiles, and the whiskers are the minimum to maximum values. For statistical analyses of MEN1 on promoter, *p* < 2.22e−16 in all comparisons. For statistical analyses of expression, *p* (Farage: DMSO vs. MI503) = 4.19e−9, *p* (K422: DMSO vs. MI503) = 3.82e-16 and *p* < 2.22e−16 in all other comparisons. **b** Violin plots showing the dependency scores of genes (*n* = 34 and 262 genes for KARPAS422 and DB cell lines, respectively) bound by MEN1 and were down-regulated by MEN1 inhibitor treatment (800 nM MI-503 for 3 days) compared to other genes (*n* = 16,820 and 16,592 genes for KARPAS422 and DB cell lines, respectively) in KARPAS422 (left) and DB cell lines (right) from DepMap screens. Lower dependency score indicates that a gene is more likely to be essential. The center line in the embedded boxplots represents the median, the box limits are the 25th and 75th percentiles, and the whiskers are the minimum to maximum values. For statistical analyses of genetic dependency in KARPAS422 and DB, *p* = 0.00236 and *p* < 2.22e−16, respectively. **c** Volcano plot showing the gene expression change after MEN1 inhibitor treatment (800 nM MI-503 for 3 days) in KARPAS422

cells (left). Genes that were also downregulated in another *EZH2*-mutated DLBCL cell line DB upon MEN1 inhibitor treatment are highlighted in black. Pathway enrichment analysis was performed on these genes (right). **d** Western blot showing total and phosphorylated S6 in Farage and KARPAS422 (K422) cells with or without the treatment of MEN1 inhibitor (800 nM MI-503 for 3 days). The experiments were repeated twice independently with similar results. **e** Bar plots showing H3K27me3 (left), MEN1/MLL1 binding (middle) signals at promoters of genes in mTORC1 signaling pathway in Farage and KARPAS422 cells. Right panel shows the corresponding gene expression (*n* = 2). **f** Heatmaps showing the fold change of MEN1 and MLL1 binding signals at mTORC1 pathway gene promoters (left), and the fold change of corresponding gene expression (right) between control (DMSO) and MEN1 inhibitor-treated (800 nM MI-503 for 3 days) *EZH2* wildtype or mutant DLBCL cell lines. **g** Relative cell counts of DLBCL cell lines treated with 1 nM Rapamycin for 96 h, normalized to the DMSO-treated controls. *n* = 3 independent experiments. Bar plots and whiskers are mean ± s.d. For statistical analyses of relative cell counts between Rapamycin-treated and control in Farage and KARPAS422, *p* = 5.20e−8 and *p* = 3.14e−5, respectively. For statistical analysis of relative cell counts of Rapamycin-treated cells between Farage and KARPAS422, *p* = 0.00431. **h** A schematic diagram showing the molecular interplay between PRC2 and MLL–MEN1 in DLBCL. Created by BioRender. For all experiments, *p* values were determined by unpaired two-tailed Student's *t*-test. **\*\****p* < 0.01; **\*\*\****p* < 0.001. Source data are provided as a Source Data file.

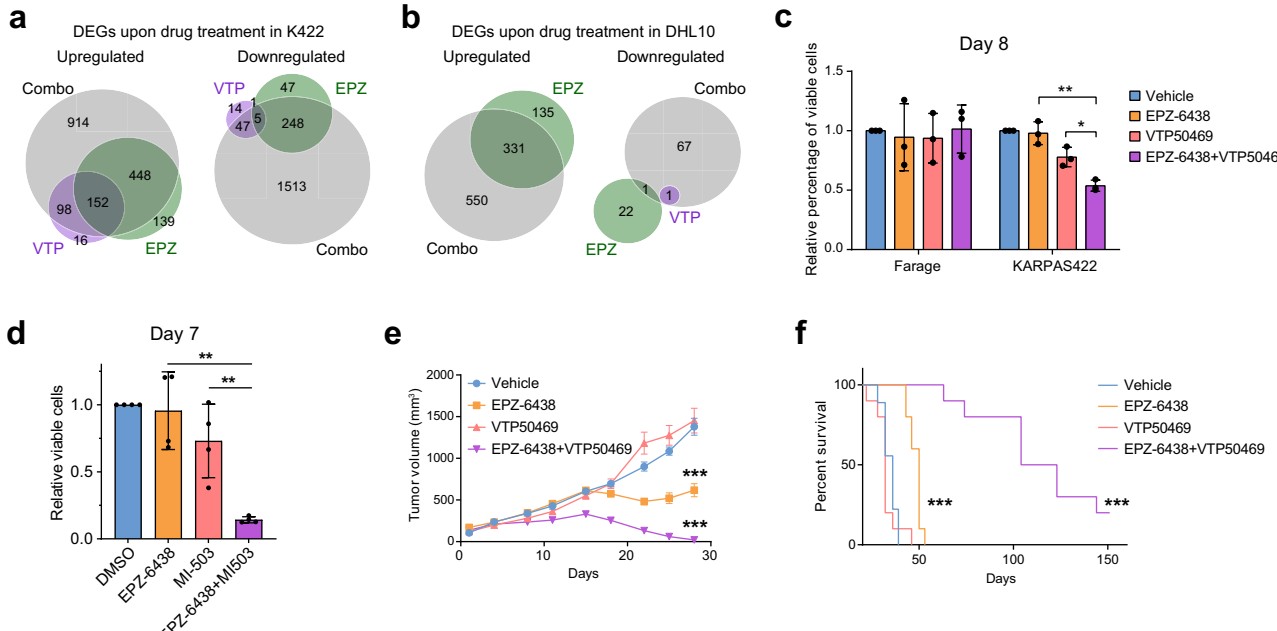

**Fig. 6 | Inhibition of both MEN1 and EZH2 in *EZH2*-mutated DLBCL. a** Venn diagram showing the overlaps of up-regulated (left) and down-regulated genes (right) between individual treatments of VTP50469 (330 nM, purple), EPZ-6438 (1 μM, green), and combo treatment of both inhibitors (gray) in KARPAS422 (K422) cells for 48 h. **b** Venn diagrams showing the overlaps of up-regulated (left) and down-regulated genes (right) between individual treatments of VTP50469 (330 nM, purple), EPZ-6438 (1 μM, green), and combo treatment of both inhibitors (gray) in SuDHL10 (DHL10) cells for 24 h. **c** Relative percentage of viable Farage or KARPAS422 cells following individual treatments of VTP50469 (160 nM, red), EPZ-6438 (200 nM, yellow), or combo treatment of both inhibitors (purple) for 8 days, normalized to the DMSO-treated controls. *n* = 3 independent experiments. Bar plots and whiskers are mean ± s.d. For statistical analyses of the relative percentage of viable cells in KARPAS422, *p* (EPZ-6438 vs. EPZ-6438 + VTP50469) = 0.00203; *p* (VTP50469 vs. EPZ-6438 + VTP50469) = 0.0115. **d** Relative percentage of viable EPZ-6438-resistant DB-r cells treated with either EPZ-6438 (1 μM, yellow) or MI-503 (800 nM, red), or both inhibitors for 7 days.

*n* = 4 independent experiments. Bar plots and whiskers are mean ± s.d. For statistical analyses of the relative percentage of viable cells, *p* (EPZ-6438 vs. EPZ-6438 + MI-503) = 0.00138; *p* (MI-503 vs. EPZ-6438 + MI-503) = 0.00522. **e** Volumes of SuDHL10 tumors treated with vehicle, VTP50469 (20 mg/kg), EPZ-6438 (125 mg/kg), or both inhibitors. *n* = 10 biologically independent animals per group. Dots and whiskers are mean ± SEM. For statistical analyses of tumor volume on Day 28, p (EPZ-6438 vs. vehicle) = 1.31e−5; *p* (EPZ-6438 + VTP50469 vs. vehicle) = 9.96e−11. **f** Kaplan–Meier survival curves of mice in each group as indicated in (**e**). P-values were determined by Log-rank (Mantel–Cox) test between each group and vehicle group. *n* = 10 biologically independent animals per group. For statistical analyses of the EPZ-6438 treated group and EPZ-6438 + VTP50469 treated group, *p* < 0.0001 for both groups when each compared to the vehicle group. For all experiments, p values were determined by unpaired two-tailed Student's *t*-test. **\****p* < 0.05; **\*\****p* < 0.01; **\*\*\****p* < 0.001. Source data are provided as a Source Data file.

efficacy of this combination therapy in vivo, we assessed combined treatment of EPZ-6438 and the orally available VTP50469[58] on tumor growth using the *EZH2* mutant SuDHL10 xenograft model. Compared to a single agent, combined treatment with EPZ-6438 and VTP50469 demonstrated markedly more potent efficacy in reducing tumor burden and extending animal survival (Fig. 6d, e). Taken together, these results suggest that the co-dependency between PRC2 and MLL–MEN1 complexes represents a potential therapeutic vulnerability for *EZH2* mutant DLBCLs.

## Discussion

In this study, we combined CRISPR-Cas9 essentiality screen datasets with molecular profiles of cancer cell lines to enable the large-scale discovery of context-defined genetic interactions between chromatin pathways. Our approach uncovered functional intra- and inter-complex interactions. We also developed a computational platform for the identification of context-specific genetic dependency by integrating various histological and molecular features including cancer type, genetic mutation, copy number variation, chromatin modification, gene expression, and drug sensitivity (https://github.com/seanchen607/deplink). Collectively, this atlas provides a resource (http://www.chaolulab-database.com/) to the research community to facilitate hypothesis generation/testing and evaluate potential chromatin-based therapeutic strategies for cancer and other human diseases.

Our DCN analysis is based on the idea that knockout of functionally related genes would produce similar phenotypic effects (e.g. fitness) across multiple conditions (e.g. cell lines), thus enabling the identification of functional interactions by measuring the fitness effects of single-gene perturbations across a panel of genetically and phenotypically heterogeneous cancer cell lines. Compared to recent studies[29–31,63–65], the current work uses distinct approaches for module calling and network visualization, and builds the pan-cancer genetic co-dependency network from the largest CRISPR-Cas9 fitness screening dataset to date (Supplementary Data 12). We observed that direct physical interactions among complex subunits were often reflected by a strong correlation (Pearson's $r > 0.4$), such as the co-dependency between C7orf26 and INTS10/13/14 ($r = 0.52$, 0.56, and 0.63, respectively). On the other hand, a moderate correlation (Pearson's $r$ between 0.34 and 0.4) may reflect an indirect interaction between two proteins functioning in the same pathway. This is the case for interactions between biochemically distinct chromatin complexes, such as MLL–MEN1 and PRC2 complex members. As proteins usually function in a complex and one complex could further functionally interact with other complexes, we came up with a complex-centric strategy using a two-step cutoff for correlation coefficient to enhance the power of discovering novel functional modules. We first use a more stringent cutoff (Pearson's $r > 0.4$) to identify complex-level interactions, and then use a less stringent cutoff (Pearson's $|r| > 0.34$) as the network cutoff to help capturing/visualizing the less strong genetic/functional interactions between members of distinct complexes. Our current pipeline has a good balance between sensitivity and specificity for predicting curated interactions in the BioGRID database, which collects experimentally validated biochemical and genetic interactions (Supplementary Fig. 10). The current co-dependency mapping approach likely lacks enough specificity to identify bona fide interactions among gene pairs with weak correlation (Pearson's $r < 0.34$), as the number of interactions rises rapidly. This limitation may be circumvented with a secondary screen using complementary approaches, such as conventional pair-wise gene interaction analysis.

A unique feature of our study is the combination of co-dependency mapping with molecular features of the cell lines analyzed (Deplink). This integrative analysis allowed us to address why a module of co-dependent genes is specifically essential to a subset of cell lines but not others, thereby inferring the potential function of the module. For example, our DCN analysis predicted that an uncharacterized gene *C7orf26* functionally interacts with *INTS10/13/14* but not genes encoding other members of the Integrator complex (module #35). This genetic interaction was validated biochemically, as C7orf26 binds to INTS10/13/14 but not the rest of the Integrator complex. These findings are consistent with recent reports that INTS10/13/14 constitute a biochemically distinct subcomplex of Integrator[64,66–68]. Our Deplink analysis shows that module #35 is selectively essential to solid tumor cell lines harboring COSMIC mutational signature 3 (failure of DNA homologous recombination repair) (Supplementary Fig. 3e). Therefore, we speculate that C7orf26/INTS10/13/14 subcomplex may participate in the DNA damage response independently of the canonical function of Integrator complex in transcriptional regulation. As another example, module #46, consisting of the INO80 chromatin remodeling complex, is required for the survival of blood cancer cell lines with histone hyperacetylation (Fig. 1d). While the causality of this correlation remains to be investigated, it is consistent with a recent study showing that INO80 complex is a central component of metabolic homeostasis that influences histone acetylation[69]. Notably, we found that a number of modules showed correlations with copy numbers and/or expression levels of module genes (Supplementary Fig. 2e, f and Supplementary Data 13, 14). We speculate that these modules represent cases of oncogene addition, where module genes are amplified/overexpressed and required for tumor cell growth. Furthermore, while currently, it is statistically underpowered to construct co-dependency networks separately for each cancer type due to the limited number of cell lines, future efforts along this line will provide key insights into cancer type-specific gene co-functionalities.

Our DCN analysis also nominates a functional co-dependency between PRC2 and MLL-MEN1, two well-characterized and transcriptionally opposing chromatin-modifying complexes. Indeed, we demonstrated that genetic or pharmacologic inhibition of MEN1 could phenocopy the effects of PRC2 knockout in inhibiting the proliferation of *EZH2* mutant and/or H3K27me3-high DLBCL and MM cells (Fig. 3). The therapeutic implication of this functional interplay between PRC2 and MLL–MEN1 is likely to extend beyond lymphoid malignancies[53,70]. For example, MLL rearrangement is a common event in pediatric leukemias, and MLL-rearranged leukemias exquisitely depend on MLL1/MEN1 for survival. In several mouse models, PRC2 has also been demonstrated to be essential for the maintenance of MLL-rearranged leukemias[71,72]. Furthermore, both MEN1 and PRC2 were identified as the top hits in a genome-wide screen for regulators of MHC-I expression as potential strategies to augment cancer immunotherapy[73]. Importantly, a combination of highly specific and potent inhibitors of PRC2 and MEN1, which are either FDA-approved or in clinical trials to treat distinct cancer types, demonstrated remarkable efficacy in the treatment of DLBCL in xenograft study (Fig. 6). These encouraging preclinical results warrant future studies to design strategies of rationally repurposing and/or combining PRC2 and MEN1 inhibitors for synergistic anti-cancer effects.

MLL–MEN1 and PRC2 complexes are the mammalian homologs of the trithorax (TrxG) and polycomb group (PcG) proteins in *Drosophila*, respectively. TrxG and PcG regulate the developmental programs and the expression of Hox genes in an antagonistic manner[74]. Consistently, we observed that regions with increased levels of H3K27me3 in *EZH2* mutant cells showed reduced MLL1 and MEN1 binding (Fig. 4). Furthermore, inhibition of EZH2 can lead to increased MLL–MEN1 binding, while inhibition of MEN1 increases H3K27me3 at surrounding regions (Supplementary Fig. 7). However, perturbing the local opposition between PRC2 and MLL–MEN1 appears to induce a complex redistribution of these epigenetic regulators at the genome-wide level, as we found that the exclusion of MLL1/MEN1 from H3K27me3-high regions promoted a re-localization and strong enrichment of these proteins at a limited number of H3K27me3-low regions, thereby elevating the expression of corresponding genes. These genes are

implicated in oncogenic pathways such as mTORC1 signaling, which has been linked to the development and drug resistance of *EZH2* mutant DLBCL[75], and their overexpression predicted a worse prognosis for DLBCL patients. Therefore, we propose that in addition to the described function of silencing B cell differentiation genes[59], *EZH2* gain-of-function mutations may also activate an MLL−MEN1-dependent oncogenic gene expression program to facilitate DLBCL pathogenesis (Fig. 5h). Notably, while this notion provides a potential mechanistic basis for the observed co-dependency between PRC2 and MLL−MEN1, it is also possible that these two complexes co-regulate the expression of bivalent genes such as MHC-I[73,76] in a cooperative manner.

In summary, our work highlights the potential of integrating parallel genetic screens with molecular phenotyping for high-throughput, context-defined genetic interaction discovery and analysis, which provides a powerful alternative to conventional biochemical approaches in the study of chromatin crosstalk. Our mechanistic investigation into the essentiality of the MLL−MEN1 complex in *EZH2*-mutated lymphomas provides one example of cancer-associated mutations in chromatin enzymes that creates a synthetic dependency on its interacting *trans*-regulatory pathway. We believe that further exploration of the resource presented here will uncover additional regulatory mechanisms involved in the progression of cancer epigenome and pave the way to new precision epigenetic therapeutic strategy.

## Methods

### Dependency correlation network analysis

The gene effect dataset of CRISPR-Cas9 essentiality screens in 1086 pan-cancer cell lines (gene effect scores derived from CRISPR knock-out screens published by Broad's Achilles and Sanger's SCORE projects, release public 2022q2)[77–79] was downloaded from Cancer Dependency Map portal (DepMap, https://depmap.org/portal/). The Pearson correlation scores were calculated for gene effect scores in pan-cancer cell lines between each two of the genes using function 'cor' in R. To validate the power of identifying complex-level interactions, human core complexes information was downloaded from CORUM database (https://mips.helmholtz-muenchen.de/corum/)[35]. The interactions mapped to reported protein interactions between CORUM core complex members and randomly selected interactions in equal numbers were generated as observed and simulated datasets, respectively. ROC curves were plotted using an R package 'pROC'. The modularity of interactions from the observed and simulated datasets was assigned as 1 and 0, respectively. The sensitivity and specificity (AUC score) of identifying CORUM complex-level interactions using different cutoffs were determined by the $p$-values from $T$-test between the correlation scores from complex-level interactions and those from global interactions. The heatmap of the Pearson correlation matrix of genes that have at least one strong interaction with other genes (matrix cutoff: Pearson's $r > 0.4$) was generated using the R package pheatmap (Pretty Heatmaps v1.0.10, parameters: clustering_method = 'ward.D', clustering_distance_cols = 'euclidean'). Based on the correlation matrix, the dependency correlation network was generated (network cutoff: Pearson's $|r| > 0.34$, only top 15 strongest co-dependency interactions for each gene were kept and singletons were removed) and visualized using GeNets (http://apps.broadinstitute.org/genets) and an R package 'geNet' (https://github.com/haneylab/geNet), which integrates a machine-learning algorithm Quack that is trained for comparing the global and local biological signal of networks and identifying the optimal network with which to interpret large genomic datasets such as cancer co-dependency relationships from project Achilles[80]. To validate our prediction of genetic interactions, previously identified genetic or physical interaction information was acquired from the BioGRID database (https://downloads.thebiogrid.org/BioGRID)[81].

### Deplink

The genetic profile (gene expression, mutations, copy number variation, and chromatin modification) and drug sensitivity (GDSC and PRISM) datasets of pan-cancer cell lines were downloaded from the DepMap portal (https://depmap.org/portal/). Association of genetic dependency with various functional characterization of cell lines was performed by a custom R package 'deplink' (https://github.com/seanchen607/deplink, parameters: cutoff.freq = 10, cutoff.percentile = 0.18, cutoff.pvalue = 0.05, cutoff.qvalue = 0.1, cutoff.diff = 0.1, cutoff.fc = 2). For each module in the dependency correlation network, Deplink selects top and bottom cell lines based on their ranking of genetic dependency of the module members and compares their molecular features (chromatin modification, gene expression, genetic mutation, copy number variation, tumor mutation burden, microsatellite instability), various signatures (COSMIC, ISG, EMT, mRNAsi, GSEA hallmark), drug sensitivity and cancer types with those of the rest cell lines. A detailed description of the code's functionality can be found on its tutorial page (https://seanchen607.github.io/deplink.html). The Stemness feature associated with oncogenic dedifferentiation was measured by mRNA stemness index (mRNAsi) which was calculated using the established approach[82]. Dependency score for each module was calculated as the opposite number of the mean value of the gene effect values of members in this module. The complete output of analysis using Deplink is provided in figshare (https://doi.org/10.6084/m9.figshare.21708425.v1) and can be queried via a searchable database (http://www.chaolulab-database.com/).

### Cell lines and cell culture

Human DLBCL cell lines KARPAS-422, DB, SuDHL-10, Farage, HBL-1, SuDHL-4 and SuDHL-5 (a gift from Jennifer E. Amengual), human leukemia cell line K562 (ATCC, #CCL-243) and human MM cell line RPMI-8226 (a gift from High-Throughput Screening Facility at the JP Sulzberger Columbia Genome Center) were maintained in RPMI1640 Medium (Gibco, #21875034) with 10% FBS (Corning, #35-010-CV). Human MM cell lines MM.1S (a gift from Selina Chen-Kiang) was maintained in RPMI1640 Medium (Gibco, #21875034) with 10% FBS (Corning, #35-010-CV), 1% MEM Non-essential amino acids (Thermo Fisher Scientific, #11140-050), 1% Penn/Strep (Thermo Fisher Scientific, #15140-122), 0.4% 1 M HEPES (Thermo Fisher Scientific, #15630-080), 1% 200 mM L-glutamine (Thermo Fisher Scientific, #25030081), 0.00035% 2-Mercaptoethanol (SIGMA, #M3148), and 0.015% 10 N NaOH (SIGMA, #SX0607N). HEK293T (ATCC, #CRL-11268) and HeLa cells (a gift from Alberto Ciccia) were maintained in DMEM GlutaMAX from Gibco with 10% FBS (Corning, #35-010-CV), and 1% Penn/Strep (Thermo Fisher Scientific, #15140-122). None of the cell lines used were authenticated. All cells were supplied with 1X Penicillin-Streptomycin (Sigma-Aldrich) and kept at 37 °C in a 5% $CO_2$ atmosphere. All cell lines were routinely tested for mycoplasma contamination.

### Lentivirus transduction

Lentivirus packaging was performed in HEK293T cells using Lipofectamine 2000 reagent (Invitrogen) in accordance with the manufacturer's instructions. The medium-containing virus was concentrated using PEG-it Virus Precipitation Solution (System Biosciences). Spin infection was performed at 1500 rpm at 33 °C for 90 min and transduced cell populations were usually selected or sorted 48 h after infection.

### Chromatin-focused CRISPR-Cas9 genetic screening and data analysis

Chromatin-focused CRISPR screening was performed as previously described[53]. Barcoded PCR products were gel purified and sequenced using the Illumina NextSeq500 instrument. FASTQ files were processed and trimmed to retrieve sgRNA target sequences using cutadapt (v4.2). Sequencing reads were aligned to the reference sgRNA

library file and counted at the gene level per sample using MAGeCK (v0.5.9.5). The beta score values were used for the final visualization.

## CRISPR-Cas9-mediated knockout

To generate the indicated knockout cell lines, cells were first transduced with LentiCas9-Blast construct (Addgene #52962) and selected with blasticidin. Guide RNAs against *MEN1* (gRNA sequences indicated in Supplementary Data 15) were cloned into pUiSEPR-puroR-RFP construct and introduced into target cells using lentiviral infection. Puromycin (1 μg/mL) was used to select gRNA+ cells.

## Functional assays

For the competitive proliferation assays using sgRNAs, the percentage of sgRNA-expressing cells (RFP+) was measured over time using flow cytometry and normalized to the starting time point (3 days after infection). Data were acquired on LSR Fortessa (BD). Flow cytometry data were analyzed using FlowJo software (V10). For proliferation assay, cell lines were plated in triplicate in 96-well plates at low density (10,000 cells per well) for each condition. Cells were treated using EZH2 inhibitor EPZ-6438 (MedChemExpress, #HY-13803) and MEN1 inhibitor MI-503 (MedChemExpress, #HY-16925) solubilized in DMSO with indicated concentration for various time points. Following drug treatment, cell viability was determined using Cell Counting Kit-8 for Cell Proliferation and Cytotoxicity Assay (Dojindo Molecular Technologies, #DJDB4000X) according to manufacturer's guidelines.

## Immunoblotting and mass spectrometry

Western blot was performed as previously described[83]. Antibodies used include anti-MEN1 (Bethyl, Cat#A300-105A, Lot#11, 1:2000 dilution), anti-MLL1 (Bethyl, Cat#A300-086A, Lot#6, 1:2000 dilution), anti-H3K27me3 (Cell Signaling, Cat#9733, Clone C36B11, Lot#19, 1:1000 dilution), anti-INTS13 (Bethyl, Cat#A303-575A, 1:1000 dilution), anti-INTS14 (Bethyl, Cat#A303-576A, 1:1000 dilution), anti-Phospho-S6 (Cell Signaling, Cat#4858, Clone D57.2.2E, 1:2000 dilution), anti-S6 (Cell Signaling, Cat#2317, Clone 54D2, 1:1000 dilution), anti-TSC1 (Cell Signaling, Cat#6935, Clone D43E2, 1:1000 dilution), anti-H3 (Abcam, Cat#ab1791, 1:10,000 dilution), and anti-Beta-actin (Abcam, Cat#ab8224, 1:2000 dilution). Antibodies were all validated by Western blot using the manufacturer's data associated with antibodies, and their authentication data. For Immunoprecipitation-based mass spectrometry, $2*10^7$ parental HeLa cells or C7orf26-FLAG-expressing HeLa cells were lysed with HS lysis buffer (50 mM Tris–HCl pH 7.9, 500 mM NaCl, 1% NP-40, 20% Glycerol, 0.5 mM PMSF, 5 mM β-mercaptoethanol) on ice for 30 min followed by centrifugation at max *g* for 20 min in 4 °C centrifuge. Soluble extracts were diluted with BC-300 buffer (20 mM Tris–HCl pH 7.9, 300 mM KCl, 20% Glycerol, 0.1 mM EDTA, 0.5 mM PMSF, 5 mM β-mercaptoethanol) and incubated with 37 μL ANTI-FLAG® M2 Affinity Gel (Sigma, Cat#A2220) overnight at 4 °C with rotation. Beads were washed five times with BC-300 buffer + 0.1% NP-40 and one time with BC-100 buffer (20 mM Tris–HCl pH 7.9, 100 mM KCl, 20% glycerol, 0.1 mM EDTA, 0.5 mM PMSF, 5 mM β-mercaptoethanol) + 0.1% NP-40. 8 M Urea in 20 mM Tris–HCl pH 8.0 was used for elution. Mass spectrometry for protein identification was performed at Columbia University Herbert Irving Comprehensive Cancer Center (HICCC) Proteomics Core. Data analysis was performed with Scaffold Proteome Software.

## Xenograft studies

Animal experiments were conducted in accordance with and with the approval of, the Columbia University Institutional Animal Care and Use Committee (IACUC). All mice were housed under specific-pathogen-free (SPF) conditions under controlled temperature (20–26 °C) and humidity (40–70%) with a 12 h/12 h light/dark cycle, following the guideline of the Columbia University animal facility. For the SuDHL-4 xenograft study, ten million SuDHL-4 cells expressing Cas9 and either

control or *MEN1* sgRNA were mixed with Matrigel and inoculated subcutaneously into the flank of 6–8 weeks old male athymic *nu/nu* mice purchased from Jackson Lab. The incidence of xenograft tumor formation was monitored and counted until animals were sacrificed at day 90 post-inoculation. For the SuDHL-10 xenograft study, 6-week-old female NOD-*scid* IL2Rgamma[null] (NSG) mice were purchased from The Jackson Laboratory. Five million SuDHL-10 cells were subcutaneously injected into the right flank. Tumor growth was measured twice a week after tumor formation. Tumor volume was calculated by volume = (width$^2$ × length)/2. Vehicle, 125 mg/kg EPZ-6438 (0.5% NaCMC and 0.1% Tween 80), 20 mg/kg VTP50469 (5% DMSO, 40% PEG400, 5% Tween 80 and 50% Saline) or combo were administrated to mice via oral gavage twice a day for 28 days. Mice were sacrificed when tumor size reached 2000 mm³. All mice were humanely euthanized by carbon dioxide inhalation, followed by cervical dislocation. Tumor growth curves and survival curves were generated with GraphPad Prism 9. Log-rank (Mantel–Cox) test was used to calculate statistical significance.

## RNA extraction and sequencing

RNA was harvested using a Trizol reagent. Illumina TruSeq RNA Sample Prep Kit (Cat#FC-122-1001) was used with 1 μg of total RNA for the construction of sequencing libraries. RNA libraries were prepared with ribosomal RNA depletion according to instructions from the manufacturer (NEB). Paired-end sequencing was performed on the Illumina NextSeq 550 sequencer.

## CUT&Tag experiment and sequencing

CUT&Tag was performed as described previously[84]. In brief, $1 × 10^5$ cells were washed with 1 ml of wash buffer (20 mM HEPES pH 7.5, 150 mM NaCl, 0.5 mM Spermidine (Sigma-Aldrich), 1× Protease inhibitor cocktail (Roche) once. Concanavalin A-coated magnetic beads (Bangs Laboratories) were washed twice with binding buffer (20 mM HEPES pH 7.5, 10 mM KCl, 1 mM MnCl₂, 1 mM CaCl₂). 10 μL/sample of beads were added to cells and incubated at room temperature for 15 min. Beads-bound cells were resuspended in 100 μL of antibody buffer (20 mM HEPES pH 7.5, 150 mM NaCl, 0.5 mM Spermidine, 0.05% Digitonin (Sigma-Aldrich), 2 mM EDTA, 0.1% BSA, 1× Protease inhibitor cocktail) and incubated with anti-MEN1 (Bethyl, Cat#A300-105A, Lot#11, 1:50 dilution), anti-MLL1 (Bethyl, Cat#A300-086A, Lot#6, 1:50 dilution), anti-H3K27me3 (Cell Signaling, Cat#9733, Clone C36B11, Lot#19, 1:100 dilution), or normal rabbit IgG (Cell Signaling, Cat#2792, Lot#9, 1:100 dilution) at 4 °C overnight on nutator. After being washed once with Dig-wash buffer (20 mM HEPES pH 7.5, 150 mM NaCl, 0.5 mM Spermidine, 0.05% Digitonin, 1× Protease inhibitor cocktail), beads-bound cells were incubated with 1 μL Guinea pig anti-rabbit secondary antibody (Antibodies Online) and 2 μL Hyperactive pA-Tn5 Transposase adapter complex made in-house in 100 μL Dig-300 buffer (20 mM HEPES pH 7.5, 0.5 mM Spermidine, 1× Protease inhibitor cocktail, 300 mM NaCl, 0.01% Digitonin) at room temperature for 1 h. Cells were washed three times with Dig-300 buffer to remove unbound antibody and Tn5 and then resuspended in 300 μL of tagmentation buffer (20 mM HEPES pH 7.5, 0.5 mM Spermidine, 1× Protease inhibitor cocktail, 300 mM NaCl, 0.01% Digitonin, 10 mM MgCl2) and incubated at 37 °C for 1 h. 10 μL of 0.5 M EDTA, 3 μL of 10% SDS, and 5 μL of 10 mg mL⁻¹ Proteinase K were added to each sample and incubated at 50 °C for 1 h to terminate tagmentation. DNA was purified using a PCR purification kit (QIAGEN) and eluted with 25 μL ddH2O. For library amplification, 21 μL of DNA was mixed with 2 μL of 10 μM Nextera i5 unique index primer (Illumina), 2 μL of 10 μM Illumina Nextera i7 unique index primer (Illumina), and 25 μL NEBNext 2× PCR mix (NEB) and subjected to the following PCR program with lid heat on in a Thermocycler: 72 °C, 5 min; 98 °C, 30 s; 13 cycles of 98 °C, 10 s and 63 °C, 10 s; 72 °C, 1 min and hold at 10 °C. To purify the PCR products, 1.1× volumes of pre-warmed Ampure XP beads (Beckman Coulter) were

added and incubated at room temperature for 10 min. Libraries were washed twice with 80% ethanol and eluted in 20 μL of 10 mM Tris–HCl, pH 8. Paired-end sequencing was performed on the Illumina NextSeq 550 sequencer.

### RNA-seq data analysis

RNA-seq reads were mapped to the human genome assembly hg38 using HISAT2 (v2.1.0). The mapped reads count of each gene was measured by featureCounts (v1.6.1). The differential gene expression was calculated and visualized by the R packages DESeq2 (v1.28.0) and ggplot2 (v3.2.1), respectively. We performed hierarchical clustering on gene expression profiles of samples using the R package pheatmap (Pretty Heatmaps v1.0.10, parameters: clustering_method = 'ward.D', clustering_distance_cols = 'euclidean', and cutree_cols = 2). Gene set enrichment analyses (GSEA) were performed by using GSEA software (v4.1.0). Pathway enrichment analysis based on The Molecular Signatures Database (MSigDB) hallmark gene set collection was performed using Enrichr (https://maayanlab.cloud/Enrichr).

### CUT&Tag data analysis

CUT&Tag reads were mapped to the human genome assembly hg38 using HISAT2 (v2.1.0). Potential PCR duplicates were removed by the function "MarkDuplicates" (parameter: REMOVE_DUPLICATES = true) of Picard (v2.23.1). Broad H3K27me3 peaks were called using SICER2 (parameters: -w 10000 -g 30000 -fdr 0.01) with IgG input as control. Narrow MEN1 and MLL1 binding peaks were called using macs2 (parameters: -q 1e-4 --max-gap 2000 --keep-dup 1) and SICER2 (parameters: -w 100 -g 300). The peaks are annotated using an R package "ChIPseeker". The overlap between called peaks was identified by the function "map" of bedtools (v2.27.1). The CUT&Tag read counts in promoter regions were measured by featureCounts (v1.6.1). Genomic enrichment of CUT&Tag signals was visualized using IGV. We performed hierarchical clustering on promoter signal enrichment of samples using the R package pheatmap (Pretty Heatmaps v1.0.10, parameters: clustering_method = 'ward.D', clustering_distance_cols = 'euclidean', and cutree_cols = 2). The signal distribution near promoter regions was measured and visualized by the functions "computeMatrix" and "plotProfile" of deepTools (v 3.3.2).

### Statistical analysis

Statistical analysis was performed using Student's *t*-test (two-sample equal variance; unpaired; two-tailed distribution) unless stated otherwise.

### Reporting summary

Further information on research design is available in the Nature Portfolio Reporting Summary linked to this article.

## Data availability

The publicly available gene effect dataset of CRISPR-Cas9 essentiality screens in 1086 pan-cancer cell lines (gene effect scores derived from CRISPR knockout screens published by Broad's Achilles and Sanger's SCORE projects, release 2022q2) used in this study is available in the Cancer Dependency Map portal (DepMap) [https://depmap.org/portal][77–79]. The publicly available human core complexes data used in this study are available in the CORUM database [https://mips.helmholtz-muenchen.de/corum][35]. The publicly available genetic or physical interaction data used in this study are available in the BioGRID database [https://downloads.thebiogrid.org/BioGRID][81]. The pan-cancer genetic co-dependency networks built from CRISPR-Cas9-based screening datasets in recent studies are publicly available in their supplementary materials and figshare [https://figshare.com/s/35a82ed1e48d0ec4e9e4][29–31,65]. The publicly available genome-wide binding data of INO80 complex and various chromatin features used in this study are available in the NCBI GEO database under accession code

GSE97411[40]. The publicly available survival information and gene expression data of a DLBCL patient cohort are available in the supplementary materials of a previous report and GDC Data Portal [https://gdc.cancer.gov/about-data/publications/DLBCL-2018][60]. The complete output of analysis using Deplink is provided in figshare [https://doi.org/10.6084/m9.figshare.21708425.v1] and can be queried via a searchable database (http://www.chaolulab-database.com). The raw sequencing data generated in this study (CUT&Tag, RNA-seq, and CRISPR-Cas9 genetic screening) are available in the NCBI GEO database under accession code GSE183487. The mass spectrometry raw data generated in this study are available in the ProteomeXchange member PRIDE database under accession code PXD033140. Source data are provided with this paper.

## Code availability

A custom R package 'deplink' dedicated to the association of genetic dependency with various cell line molecular features, as well as example data are accessible via GitHub at the following address: https://github.com/seanchen607/deplink, including a readme.txt file providing instructions for installing and running the software. A detailed tutorial for software's installing, running, and the expected output is exhibited on a GitHub page (https://seanchen607.github.io/deplink.html).

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

## Acknowledgements

We thank Giuseppe Leuzzi, Lei Ding, Liling Wan, and Tahir Sheikh for technical help, and William Seong for experimental support. A schematic diagram of the current model was created using BioRender. These studies were supported by the Columbia Velocity Award (to J.E.A. and C.L.), the Pew-Stewart Scholars for Cancer Research (to C.L.), the V Scholar Grant funded by the V Foundation (to C.L.), and NIH grants (R35GM138181 and R01DE031873 to C.L.; R01CA197774 to A.C.; R01CA172492 to L.P.). F.J.S.R. is an HHMI Hanna Gray Fellow and was supported by the V Foundation for Cancer Research (V2022-028), NCI Cancer Center Support Grant P30-CA1405, the Ludwig Center at MIT (2036636), Koch Institute Frontier Awards (2036648 and 2036642), and the MIT Research Support Committee (3189800). These studies used the resources of the Columbia flow cytometry shared resource and the Judith P. Sulzberger Columbia Genome Center, funded in part through NCI Cancer Center Support Grant P30CA013696.

## Author contributions

Conceptualization, X.C., A.C., and C.L.; Methodology, X.C., A.C., and C.L.; Formal Analyses, X.C., Y.L., X.X., F.Z., B.E., M.A.P., J.T.M., D.K., V.S., M.M., C.S., F.J.S.R., Y.M.S.; Investigation, X.C.; Writing—original draft, X.C. and C.L.; Writing—review & editing, X.C., Y.L., F.Z., D.K., A.C., J.E.A., L.P., and C.L.; Supervision, A.C., J.E.A., L.P., and C.L.; Funding acquisition, A.C., J.E.A., L.P., F.J.S.R., and C.L.

## Competing interests

The authors declare no competing interests.

## Additional information

[1]Department of Genetics and Development, Columbia University Irving Medical Center, New York, NY 10032, USA. [2]Union Hospital Cancer Center, Tongji Medical College, Huazhong University of Science and Technology, 430022 Wuhan, China. [3]Division of Hematology and Oncology, Department of Medicine, Columbia University Irving Medical Center, New York, NY 10032, USA. [4]Institute for Cancer Genetics, Columbia University Irving Medical Center, New York, NY 10032, USA. [5]Department of Pathology and Cell Biology, Columbia University Irving Medical Center, New York, NY 10032, USA. [6]David H. Koch Institute for Integrative Cancer Research, Massachusetts Institute of Technology, Cambridge, MA 02142, USA. [7]Department of Biology, Massachusetts Institute of Technology, Cambridge, MA 02142, USA. [8]Herbert Irving Comprehensive Cancer Center, Columbia University Irving Medical Center, New York, NY 10032, USA. [9]Present address: Marine College, Shandong University, 264209 Weihai, China. [10]These authors contributed equally: Xiao Chen, Yinglu Li. ✉e-mail: cl3684@cumc.columbia.edu

