## [Peer Review File · Nature Communications]

Context-defined cancer co-dependency mapping identifies a functional interplay between PRC2 and MLL-MEN1 complex in lymphomaReviewers' Comments:

Reviewer #1:

Remarks to the Author:

In this manuscript, Chen et al. claimed to develop an analysis algorithm to describe a co-dependency gene map using publicly available DEPMAP CRISPR screen data. We identify 84 co-dependency modules by incorporating cell line mutational, epigenome, gene expression, and drug sensitivity profiles. They further picked an example of the PRC2 and MLL-MEN1 complex and showed that H3K27me3 regulates the genome-wide distribution of MLL1 and MEN1. EZH2 gain-of-function mutations displayed hypersensitivity of pharmacologic inhibition of MEN1 in lymphoma cell lines. Overall, lack of novelty is the major issue as it stands right now. Many studies were conducted to re-analyze the rich publicly available datasets released from the DEPMAP consortium and to identify the dependency and co-dependency by using different algorithms or comparisons. Most of the findings and selected examples have been reported elsewhere (MENi, KAT7, etc.). However, I think the game changer of using "DEPMAP data" is optimizing the condition and improving data quality. For instance, the original whole-genome CRISPR library used the CRISPR dropout screens of DEPMAP based on "predicted sgRNAs" other than validated sgRNAs. Further efforts by re-screening with more efficient sgRNAs, sub-libraries, and dropout criteria will certainly help to identify the overlooked candidates. From this perspective, simply re-analyzing the existing data will not significantly polish the readout of current DEPMAP data. Not surprisingly, most "co-dependency modules" are well-known. In addition, there are significantly incremental concerns about the uncertain quality control, data interpretation, and lack of *in vivo* functional validation, which dampened my enthusiasm to recommend for publication at the current status. I hope that the authors will find the enclosed comments helpful.

1. The most concern is about the novelty and justification of the current DCN and Deplink compared with other analysis pipelines or algorithms. Taking the publicly available DEPMAP datasets and predicting ~80 modules and ending with testing one or two really does not exhibit the advantage and value of the current study. If the author can systematically compare and justify their findings (Fig 1-3), they may want to submit to other journals with specific interests. Or they can spend more effort to validate the biology of the novel candidate and pursue therapeutic advantage.
2. Although DEPMAP data are collected from cell lines, they were used to predict the benefit of applying findings to PDX or primary tumors by *in vivo* models, which are hard to edit. All of the validation and drug tests were performed by *in vitro* culture (MTT, competition). The only transplantation model is from the cell line as well. A mouse model or PDX system is required to strengthen the findings.
3. The molecular mechanism underlying EZH2 mutant and MENi sensitivity were quite at an early phase.
4. HBO/KAT7 dependency in MLL-r has been well reported previously in both human and mouse systems (Elife. 2021 Aug 25;10:e65872.; Leukemia. 2021 Apr;35(4):1012-1022 and Nature. 2020 Jan;577(7789):266-270.)
5. The conclusion of H3K27ac3 domain in EZH2-mutant cells derives redistribution of MLL-MEN1 complex requires more likely a correlation. Functional and dynamic link should be addressed using acute depletion of EZH2/H3K27me3 system followed with characterizing MLL1-MEN1 binding dynamics particularly at early phase upon perturbation.

Reviewer #2:

Remarks to the Author:

In this study, the authors used DepMap data to build a co-dependency network to help identify novel factors that cooperate together for maintaining survival across diverse cell types. They then created their own analysis pipeline, Deplink, to link up these dependencies with cancer subtypes, drug

dependencies, chromatin profiles, gene mutations and gene expression profiles. They validated their analyses using several approaches including highlighting the identification of known factors as well as using IP/westerns to validate a predicted interaction between C7orf26 and several integrator proteins.

They went on to concentrate on a module that contains KMT2A, MEN1 and several PRC2 complex components. Further analysis revealed that B cell lymphoma cell lines were particularly sensitive to loss of these factors, and function cooperatively. EZH2 gain of function mutant KARPAS422 cells had expanded H3K27me3 domains as well as reduced KMT2A and MEN1 binding profiles. They go on to show that EZH2 mutant cells are particularly sensitive to MEN1 inhibitors and that combined EZH2/MEN1 inhibition reduces tumour growth in a xenograft model.

This is a well written and clearly explained paper and I only have a few comments. There are a few details of the model and observed interactions that are not entirely clear.

1. What happens if you treat KARPAS422 cells with EZH2 inhibitors? Does this decrease H3K27me3 levels and expand KMT2A/MEN1 binding profiles? The model in Fig 6i might predict this.
2. What happens if you treat Farage cells with MEN1 inhibitors (the authors may want to consider the much more potent Syndax inhibitor that they use in Figure 6g rather than the weak MI-503), does this alter EZH2/H3K27me3 profiles? Or does the inhibition of binding only work in one direction?
3. Is there a reason why this is a B cell lymphoma specific effect? Does this dependency only exist where there is hyperactive EZH2? Does EZH2 inhibition in other cancer cell types (e.g. AML, ALL, others) cause a redistribution of KMT2A/MEN1 binding? Does KMT2A/MEN1 binding have reduced binding in all cell lines with higher levels of H3K27me3?

Reviewer #3:

Remarks to the Author:

In this paper Lu et al. described a method for mapping the co-dependency using CRISPR-Cas9 KO fitness screens in order to quantify the gene-gene functional relationships. Modules identified in the co-essentiality networks are later associated with mutational, epigenome, gene expression and drug sensitivity of the cell lines in order to determine the mechanisms underlying essentiality of these modules. The results of this mapping and association can be used for discovering new trans-regulatory interactions as well as new targets for synthetic lethality. Using this resource authors show a surprising co-dependency between PRC2 and MLL-MEN1 and validated the functional relationship.

The paper describes an approach for identification of functionally similar genes (modules) using the co-essentiality networks. Although this is an interesting and promising idea, using the co-essentiality networks for identification of functions of uncharacterised genes has been used in many papers and recently a genome-wide study looked at the co-essential modules in this network for similar purposes. Here the main addition is the association of the modules with the genomic and epigenetic features of the cell lines (Deplink). Here are some small questions/comments related to the generation of the co-essentiality networks and their association with cell line features:

- The co-essentiality network is generated using 689 pan-cancer cell lines from Broad institute. There is a similar dataset from Sanger Institute as well as a combined data-set (published recently) where two datasets are normalised/bias removed and combined together. It would be important to generate the co-essentiality networks for the merged dataset. The combined data-set should give the most accurate representation of co-essentiality in these screens (better bias removal and more samples)
- In a related note it would be possible to compare the accuracy and robustness of these networks using the CORUM core complex member. It would be also nice to look at hu.MAP, a mass spectrometry (MS) protein-protein interaction database and STRING protein-protein interaction database for physical interactions only. The reliability of the co-essentiality network is quite important

for the down-stream analysis.

- Some of the essentiality modules are associated with some cancer types. It would be interesting to generate these co-essentiality networks separately for each cancer type and look at the associations within these networks. Demonstrating the same functional similarity in the related cancer types can be important and maybe similar associations can be identified in other cancer types.
- While generating the co-essentiality networks only a subset of the genes (6464 genes) are selected. I think this is not justified quite well in the paper. I think using only a small portion of the genes for generating these networks is quite limiting for discovery purposes. Recently published co-essentiality papers all focus on the genome wide essentiality modules. Selecting a subset of the genes will affect the topology of the network quite a bit as well as the statistical analysis of the association of these modules with cell line features (number of modules will be different).
- Identification of the essentiality modules is pretty basic. Only genes that are co-essential with some correlation coefficient above certain threshold are used. Here also the subset of the genes play an important role. With a small number of genes, it is possible to get disconnected modules however in the genome wide analysis it will be harder to cluster the essentiality modules. However there are wide range of computational methods for detecting modules in networks (especially there are a lot of methods for co-expression networks) Analysing this data using the whole genome and some of these module detection methods would be more interesting and novel. This will also increase the scope and usefulness of the results as a resource for discovery.
- The essentiality networks are associated with cell line features. Although mutational signatures and pathway expressions are used for associations, genetic variants such as SNPs, Indels and copy number are not used. It could be interesting to look at associations for specific variants or expression of individual genes.
- Deplink pipeline is not really well documented and seems to be pretty straightforward (for testing associations statistically). It will be nice to have more clarification about the novelty of this analysis.
- The results of the paper is pretty good and interesting. Identification of the new co-dependencies and their validation is quite interesting and important. The paper is written well and described the methods and results clearly.

We thank the editor and reviewers for their valuable comments, which allowed us to substantially improve our manuscript. To address the reviewers' comments and strengthen our main conclusions, we performed several major experiments that have been added to the revised version of the manuscript:

- **Enhance the novelty, impact and robustness of Deplink analysis pipeline:**

Our Deplink analysis (ie. identification of the context underlying module essentiality) represents a new and powerful approach to enable the discovery of gene co-functionality as well as context-specific synthetic lethal interactions in cancer. We have revised the title and figure structure (consolidate Figures 1-3 into Figure 1) to focus on this novelty of our work. We have also added discussion (Line 457-460) to differentiate our approach and findings to previous work. Furthermore, we have created a database website (<http://www.chaolulab-database.com/>) to make our Deplink analysis results more accessible to readers. Finally, we have compared the list of our co-dependency modules to that of other analysis pipelines (Supplementary Table 12), as well as to physical interaction databases including CORUM and BioGRID (Supplementary Fig. 1a, Supplementary Table 1) to demonstrate the robustness of our analysis.

- **Improve the computational analysis pipeline:**

We re-generated the co-dependency network using a combined CRISPR knockout screen dataset derived from the combination of Broad's Achilles and Sanger's SCORE projects, and using all 16,854 non-essential genes (revised workflow in Fig. 1a). In addition, for the module detection from co-dependency network, we used a machine-learning algorithm Quack (integrated in GeNets). While our major findings are not affected, these efforts significantly improved the network by increasing the number of our co-dependency modules from 84 to 145. We also made a tutorial page to explain our code's functionality in detail (<https://seanhen607.github.io/deplink.html>).

- **Demonstrate a causal and bi-directional role of H3K27me3 in opposing MLL1/MEN1 binding:**

We performed CUT&Tag experiments with H3K27me3, MLL1 and MEN1 in *EZH2* mutant DLBCL line (Karpas422) upon EZH2 inhibitor (EPZ-6438) treatment. EZH2 inhibition effectively abolished genome-wide H3K27me3 enrichment, and regained MLL1/MEN1 bindings at previously H3K27me3-high regions. Concomitantly, we observed decreased signal abundance of pre-existing MLL1/MEN1 peaks (Fig. 4e, Supplemental Fig. 7a-b). We also assessed and observed a gain of H3K27me3 surrounding MLL1/MEN1 binding peaks following MEN1 inhibitor treatment in both KARPAS422 and Farage cells (Supplemental Fig. 7c-e). These results suggest that the opposition between MLL-MEN1 complex and H3K27me3 is direct and bidirectional.

- **Further characterize the molecular mechanism underlying EZH2 mutant DLBCL and MEN1 sensitivity:**

We have added three major investigations to this end: (1) we performed independent chromatin-focused sub-library screens to unbiasedly show the preferential dependency of MLL/MEN1 in *EZH2* mutant DLBCL (Fig. 3b-c); (2) We demonstrate that the co-dependency between PRC2 and MLL-MEN1, and H3K27me3-driven redistribution of MLL-MEN1 complex binding, apply more broadly to hematopoietic tumors with high levels of H3K27me3 beyond B-cell lymphomas such as multiple myelomas (Supplemental Fig. 6); (3) We demonstrate that pharmacologic inhibition of mTORC1 by rapamycin phenocopied MEN1 inhibitor's preferential toxicity towards KARPAS422 cells (Fig. 5g); while knockout of *TSC1*, an upstream negative regulator of mTORC1 signaling, partially rescued the decreased viability of KARPAS422 cells following MEN1 inhibition (Supplemental Fig. 8f). These results support a functional role of MEN1-regulated pathways such as mTORC1 signaling in *EZH2* mutant DLBCL cells' hypersensitivity to MEN1 inhibition.

Point-by-point response to reviewer's comments:

Reviewer #1:

In this manuscript, Chen et al. claimed to develop an analysis algorithm to describe a co-dependency gene map using publicly available DEPMap CRISPR screen data. We identify 84 co-dependency modules by incorporating cell line mutational, epigenome, gene expression, and drug sensitivity profiles. They further picked an example of the PRC2 and MLL-MEN1 complex and showed that H3K27me3 regulates the genome-wide distribution of MLL1 and MEN1. EZH2 gain-of-function mutations displayed hypersensitivity of pharmacologic inhibition of MEN1 in lymphoma cell lines. Overall, lack of novelty is the major issue as it stands right now. Many studies were conducted to re-analyze the rich publicly available datasets released from the DEPMap consortium and to identify the dependency and co-dependency by using different algorithms or comparisons. Most of the findings and selected examples have been reported elsewhere (MENi, KAT7, etc.). However, I think the game changer of using “DEPMap data” is optimizing the condition and improving data quality. For instance, the original whole-genome CRISPR library used the CRISPR dropout screens of DEPMap based on “predicted sgRNAs” other than validated sgRNAs. Further efforts by re-screening with more efficient sgRNAs, sub-libraries, and dropout criteria will certainly help to identify the overlooked candidates. From this perspective, simply re-analyzing the existing data will not significantly polish the readout of current DEPMap data. Not surprisingly, most “co-dependency modules” are well-known. In addition, there are significantly incremental concerns about the uncertain quality control, data interpretation, and lack of in vivo functional validation, which dampened my enthusiasm to recommend for publication at the current status. I hope that the authors will find the enclosed comments helpful.

We thank the reviewer for the helpful comments. Here, the reviewer raised two issues. First, we agree with the reviewer that previous studies have performed similar analysis to identify genetic co-dependency using DepMap datasets (see Reference 28,29,62,63 in the manuscript). However, a key novelty of our analysis (ie. Deplink) is to associate co-dependency modules with specific phenotypic and molecular features of cancer cell lines (**Fig. 1, Supplementary Fig. 2**). To our knowledge, only one recent publication has correlated co-dependency modules to cancer tissue types (Wainberg et al., *Nat Genetics* 2022). Our Deplink analysis, in contrast, is much more comprehensive and results in the identification of co-dependency modules that are specific to cancer mutational signatures, gene expression signatures, chromatin modifications and drug sensitivity. As such, our analysis reveals new insights not reported elsewhere, including the predicted synthetic lethal interactions between *PRPF39-TRNAU1AP* gene pair in mismatch repair-deficient cancer cells, and MLL-MEN1 complex in H3K27me3-high blood cancers (which we extensively validated and investigated) – just to name a few.

At the reviewer's suggestion to focus on the novelty of our work, we revised the title, and moved results related to the identification of co-dependency modules to Supplementary Fig. 1, with main Fig. 1 now exclusively describes the Deplink analysis. Furthermore, we have created a database website (<http://www.chaolulab-database.com/>) to make our Deplink analysis results more accessible to readers. We also note that, with the addition of a significant amount of new data, the majority of the revised manuscript (Figures 2-6) report original findings about the functional validation and mechanistic characterization of the co-dependency between MLL-MEN1 and PRC2 complex in lymphomas.

Secondly, the reviewer made the excellent suggestion of performing re-screens with independent sub-libraries to complement the analysis of DepMap data. To this end, we employed an independently designed, chromatin-focused CRISPR gRNA library and performed a drop-out screen in *EZH2*-mutant KARPAS422 cells. We found a high concordance between the results of our screen and DepMap screen ($r=0.7557$, **Fig. 3b**). Furthermore, our chromatin sub-library screens showed that the MLL-MEN1 and PRC2 co-dependency module (module #8) was significantly more essential to *EZH2*-mutant

KARPAS422 cells but not *EZH2*-wildtype K562 cells (Fig. 3c, Supplementary Fig. 4b), consistent with DepMap analysis result.

1. The most concern is about the novelty and justification of the current DCN and Deplink compared with other analysis pipelines or algorithms. Taking the publicly available DEPMAP datasets and predicting ~80 modules and ending with testing one or two really does not exhibit the advantage and value of the current study. If the author can systematically compare and justify their findings (Fig 1-3), they may want to submit to other journals with specific interests. Or they can spend more effort to validate the biology of the novel candidate and pursue therapeutic advantage.

As described above, while our DCN analysis (ie. identification of co-dependency modules) overlaps with previous efforts, the Deplink analysis (ie. identification of the context underlying module essentiality) represents a new and powerful approach to enable the discovery of gene co-functionality as well as context-specific synthetic lethal interactions in cancer. In response to the reviewer's comment, we have revised the structure of figures (consolidate Figures 1-3 into Figure 1) to emphasize this point better. We have also added discussion (Line 457-460) to differentiate our approach and findings to previous work.

In parallel, we have spent major effort to generate more mechanistic insights and explore further therapeutic implications for the co-dependency between MLL-MEN1 and PRC2 in lymphomas. The mechanism-related data are now included in Figures 4 and 5 and the details can be found in our response to the reviewer's point #3. Therapeutics-related data are now part of Figures 3 and 6 and the details can be found in our response to the reviewer's point #2. Together, we have added **29** new figure panels to expand and strengthen our main conclusions on this part.

2. Although DEPMAP data are collected from cell lines, they were used to predict the benefit of applying findings to PDX or primary tumors by in vivo models, which are hard to edit. All of the validation and drug tests were performed by in vitro culture (MTT, competition). The only transplantation model is from the cell line as well. A mouse model or PDX system is required to strengthen the findings.

While we agree with the reviewer that our findings would be strengthened with the addition of PDX system, currently there is only one well-characterized DLBCL PDX panel for preclinical agent evaluation (offered by Crown Bioscience), and unfortunately there is no *EZH2* gain-of-function mutant PDX line in this panel. Given the difficulty to access relevant PDX system, we have instead performed the following experiments to strengthen our conclusions:

- We demonstrate the synergy between EZH2 inhibitor and MEN1 inhibitor in additional DLBCL cell lines, showing that combined inhibition of EZH2 and MEN1 had significantly larger impact on the cancer cell transcriptome and viability in *EZH2* mutant (KARPAS422 and SuDHL10) but not *EZH2* wildtype (Farage) DLBCL lines (**Fig. 6a-c, Supplementary Fig. 9a-c**).

- We performed survival analysis of the SuDHL10 xenograft study and found that combined inhibition of EZH2 and MEN1 markedly extended the animal survival, with ~25% of the mice remained tumor free >150 days following tumor implantation (**Fig. 6f**).

- We expanded the co-dependency between MLL-MEN1 and PRC2 beyond DLBCL to multiple myeloma (MM) (**Supplementary Fig. 4a**). We show that MEN1 inhibitor, alone or in combination with EZH2 inhibitor, was significantly more toxic to MM cell line that shows high levels of H3K27me3 (RPMI-8226) compared to H3K27me3-normal MM line (MM.1S) (**Fig. 3g, Supplementary Fig. 9d**).

Fig. 6: (a) Venn diagrams showing the overlaps of up-regulated (left) and down-regulated genes (right) between individual treatments of VTP50469 (330 nM, purple), EPZ-6438 (1 μ M, green), and combo treatment of both inhibitors (grey) in KARPAS422 (K422) cells for 48 hr.

(b) Venn diagrams showing the overlaps of up-regulated (left) and down-regulated genes (right) between individual treatments of VTP50469 (330 nM, purple), EPZ-6438 (1 μ M, green), and combo treatment of both inhibitors (grey) in SuDHL10 (DHL10) cells for 24 hr.

(c) Relative percentage of viable Farage or KARPAS422 cells following individual treatments of VTP50469 (160 nM, red), EPZ-6438 (200 nM, yellow), or combo treatment of both inhibitors (purple) for 8 days, normalized to the DMSO-treated controls (n=3). Bar plots and whiskers are mean \pm s.d. P-values were determined by two-tailed Student's t-test. *, p < 0.05.

Supplementary Fig. 9: (a) Pathway annotation of differentially expressed genes (DEGs) in KARPAS422 (K422) cells upon combo treatment of both MEN1 and EZH2 inhibitors.

(b) Pathway annotation of differentially expressed genes (DEGs) in SuDHL10 (DHL10) cells upon combo treatment of both MEN1 and EZH2 inhibitors.

(c) Relative percentage of viable SuDHL10 cells following individual treatments of VTP50469 (160 nM, red), EPZ-6438 (200 nM, yellow), or combo treatment of both inhibitors (purple) for 8 days, normalized to the DMSO-treated controls (n=3). Bar plots and whiskers are mean \pm s.d. P-values were determined by two-tailed Student's t-test. *, $p < 0.05$; **, $p < 0.01$.

Fig. 6f: Kaplan-Meier survival curves of mice in each group as indicated in (e). P-values were determined by Log-rank (Mantel-Cox) test between each group and vehicle group. ***, $p < 0.001$.

Supplementary Fig. 4a: Dot plot showing the dependency landscape of module #8 in cell lines across cancer types.

Significant enrichments are denoted by asterisks (*, $p < 0.05$; ***, $p < 0.001$). Mean dependency scores are denoted by red bars.

(Left) Fig. 3g: Relative percentage of viable cells after 4-day treatment of MEN1 inhibitor VTP50469 with indicated dosage (n=3) for H3K27me3-high (RPMI-8226) and H3K27me3-low (MM.1S) multiple myeloma cell lines. Dots and whiskers are mean \pm s.d.

(Right) Supplementary Fig. 9d: Relative percentage of viable MM.1S or RPMI-8226 cells following individual treatments of VTP50469 (160 nM, red), EPZ-6438 (200 nM, yellow), or combo treatment of both inhibitors (purple) for 4 days, normalized to the DMSO-treated controls (n=3). Bar plots and whiskers are mean \pm s.d. P-values were determined by two-tailed Student's t-test. *, p < 0.05.

3. The molecular mechanism underlying EZH2 mutant and MEN1 sensitivity were quite at an early phase.

We thank the reviewer for this constructive feedback. Our mechanistic model (**Fig. 5h**) is that in *EZH2* mutant DLBCL cells, the expansion of H3K27me3 domains restricts and concentrates the binding of MLL-MEN1 complex to a small number of accessible gene promoters, which in turn drives a higher and MEN1-dependent expression of genes involved in key oncogenic pathways (eg. mTORC1 signaling) that are required for tumor growth. To improve the breadth and depth of our mechanistic studies, we have included four additional lines of investigation:

- We demonstrate that similar mechanism can be applied to cancer types beyond DLBCL such as multiple myelomas (MM). Specifically, we performed CUT&Tag profiling of H3K27me3-high RPMI-8226 and H3K27me3-normal MM.1S cells. We found an increased number of H3K27me3 large domains in RPMI-8226 cells, which was accompanied by a decrease in the number of MEN1 peaks (**Supplemental Fig. 6a-b**). Importantly, both genome-wide analysis and inspection at representative loci revealed higher signal abundance for MEN1/MLL1 promoter peaks in RPMI-8226 cells (**Supplemental Fig. 6c-d**), mirroring that in KARPAS422 cells.
- We show that higher levels of MLL1/MEN1 binding in H3K27me3-high DLBCL or MM cells can be reversed with depletion of H3K27me3. Please see comments to Point #6 for more details.
- We assessed H3K27me3 enrichment in Farage and KARPAS422 cells before and after the treatment of MEN1 inhibitors MI-503 and VTP50469 using CUT&Tag. We observed a gain of H3K27me3 surrounding MLL1/MEN1 binding peaks following MEN1 inhibitor treatment in both KARPAS422 and Farage cells (**Supplemental Fig. 7c-e**), suggesting that the opposition between MLL-MEN1 complex and H3K27me3 is bidirectional.
- We performed additional analysis to test if MEN1-dependent genes (eg. mTORC1 signaling pathway) are more essential to the fitness of *EZH2* mutant DLBCL cell lines. First, we show that genes that were downregulated following MEN1 inhibition showed significantly lower dependency scores in genome-wide knockout screens of KARPAS422 and DB cells compared to other genes (**Fig. 5b**). Second, we show that pharmacologic inhibition of mTORC1 by rapamycin phenocopied MEN1 inhibitor's preferential toxicity towards KARPAS422 cells (**Fig. 5g**). Furthermore, knockout of *TSC1*, an upstream negative regulator of mTORC1 signaling, partially rescued the decreased viability of KARPAS422 cells following MEN1 inhibition (**Supplemental Fig. 8f**). These results support a functional role of MEN1-regulated pathways such as mTORC1 signaling in *EZH2* mutant DLBCL cells' hypersensitivity to MEN1 inhibition.

Supplementary Fig. 6: (a) Venn diagram showing the genomic partition (Mb) of H3K27me3 large domains (> 100 kb) shared between H3K27me3-high MM cell line RPMI-8226 (purple) and H3K27me3-low MM cell line MM.1S (green). (b) Venn diagrams showing the MEN1 (left) or MLL1 (right) binding promoters shared between RPMI-8226 (purple) and MM.1S (green) cells. (c) Violin plots showing the normalized signal abundance (Z score) of MEN1 (left) and MLL1 (right) peaks at all MEN1/MLL1-bound promoters in MM.1S and RPMI-8226 cells. EPZ, EZH2 inhibitor EPZ-6438. The center line in the embedded boxplots represents the median, the box limits are the 25th and 75th percentiles, and the whiskers are the minimum to maximum values. P-values were determined by two-tailed Student's t-test. ***, $p < 0.001$; ns, not significant. (d) Integrative Genomics Viewer snapshot showing the landscape of H3K27me3, MEN1 and MLL1 binding at the LRR34 (left), LRR26 (middle) and LAMTOR4 (right) gene loci in MM cell lines. EPZ, EZH2 inhibitor EPZ-6438.

Supplementary Fig. 7: (c) Heatmaps showing levels of MEN1/MLL1 binding and H3K27me3 around MEN1 binding peaks upon VTP-50469 treatment in Farage (left) and KARPAS422 cells (right). Each row represents one MEN1 binding peak and shows CUT&Tag signal intensity (CPM) within 5 kb range on each side of the peak center.

(d) Violin plots showing the normalized signal abundance (CPM) of MEN1 (left) and MLL1 (middle) peaks, and H3K27me3 (right) at all MEN1-bound promoters (within 5 kb range on each side of the promoter center) in Farage and KARPAS422 cells. VTP, MEN1 inhibitor VTP-50469. The center line in the embedded boxplots represents the median, the box limits are the 25th and 75th percentiles, and the whiskers are the minimum to maximum values. P-values were determined by two-tailed Student's t-test. ***, $p < 0.001$.

(e) Integrative Genomics Viewer snapshot showing the landscape of H3K27me3, MEN1 and MLL1 binding at the H4C11 (left), TDP2 (right) gene loci in Farage and KARPAS422 cells treated with VTP-50469 or DMSO.

(Left) Fig. 5b: Violin plots showing the dependency scores of genes bound by MEN1 and were down-regulated by MEN1 inhibitor treatment (800 nM MI-503 for 3 days) compared to other genes in KARPAS422 (left) and DB cell lines (right) from DepMap screens. Lower dependency score indicates that a gene is more likely to be essential. The center line in the embedded boxplots represents the median, the box limits are the 25th and 75th percentiles, and the whiskers are the minimum to maximum values. P-values were determined by two-tailed Student's t-test. ***, $p < 0.001$.

(Middle) Fig. 5g: Relative cell counts of DLBCL cell lines treated with 1 nM Rapamycin for 96 hours, normalized to the DMSO-treated controls. P-values were determined by two-tailed Student's t-test. *, $p < 0.05$; ***, $p < 0.001$.

(Right) Supplementary Fig. 8f: Left, Western blot showing the knockout of TSC1 in KARPAS422 cells using sgRNAs. Right, relative percentage of viable KARPAS422 (K422) cells with or without TSC1 knockout under VTP50469 treatment in different concentrations for 8 days, normalized to the DMSO-treated controls (n=3). Bar plots and whiskers are mean \pm s.d. P-values were determined by two-tailed Student's t-test. *, $p < 0.05$; **, $p < 0.01$; ns, not significant.

4. HBO/KAT7 dependency in MLL-r has been well reported previously in both human and mouse systems (Elife. 2021 Aug 25;10:e65872.; Leukemia. 2021 Apr;35(4):1012-1022 and Nature. 2020 Jan;577(7789):266-270.)

At the reviewer's comment, we have removed this Supplementary Figure from the revised manuscript to focus on our novel findings.

5. The conclusion of H3K27ac3 domain in EZH2-mutant cells derives redistribution of MLL-MEN1 complex requires more likely a correlation. Functional and dynamic link should be addressed using acute depletion of EZH2/H3K27me3 system followed with characterizing MLL1-MEN1 binding dynamics particularly at early phase upon perturbation.

We thank the reviewer for this excellent suggestion. We performed H3K27me3 CUT&Tag experiments with spike-in controls for *EZH2* mutant DLBCL line (Karpas422) or H3K27me3-high MM line (RPMI-8226) upon *EZH2* inhibitor (EPZ-6438) treatment at an early time point (72 hours). *EZH2* inhibition effectively abolished genome-wide H3K27me3 enrichment, and regained MLL1/MEN1 bindings at previously H3K27me3-high regions (**Supplemental Fig. 7a-b**). Concomitantly, we observed decreased signal abundance of pre-existing MLL1/MEN1 peaks (**Fig. 4e, Supplemental Fig. 7a-b**). A similar "titration" of MLL1/MEN1 binding was also observed in RPMI-8226 cells following EPZ-6438 treatment

(Supplemental Fig. 6c). Together, these results suggest a causal function of H3K27me3 in regulating genome-wide patterns of MLL1/MEN1 binding.

Supplementary Fig. 7: (a) Heatmaps showing levels of MEN1/MLL1 binding and H3K27me3 around gained (top), shared (middle) and lost (bottom) MEN1 binding peaks upon EPZ-6438 treatment in KARPAS422 cells. Each row represents one MEN1 binding peak and shows CUT&Tag signal intensity (CPM) within 5 kb range on each side of the peak center.

(b) Integrative Genomics Viewer snapshot showing the landscape of H3K27me3, MEN1 and MLL1 binding at the CD24 (left), DDX5 (middle) and CDRT15P3 (right) gene loci in KARPAS422 cell treated with EPZ-6438 or DMSO.

(Left) Fig. 4e: Violin plots showing the normalized signal abundance (Z score) of MEN1 (left) and MLL1 (right) peaks at all MEN1/MLL1-bound promoters in Farage and KARPAS422 cells (all), or only those MEN1/MLL1-bound promoters shared between Farage and KARPAS422 cells (shared). EPZ, EZH2 inhibitor treatment using 1 μM EPZ-6438 for 72 hr. The center line in the embedded boxplots represents the median, the box limits are the 25th and 75th percentiles, and the whiskers are the minimum to maximum values. P-values were determined by two-tailed Student's t-test. ***, $p < 0.001$.

(Right) Supplementary Fig. 6c: Violin plots showing the normalized signal abundance (Z score) of MEN1 (left) and MLL1 (right) peaks at all MEN1/MLL1-bound promoters in MM.1S and RPMI-8226 cells. EPZ, EZH2 inhibitor EPZ-6438. The center line in the embedded boxplots represents the median, the box limits are the 25th and 75th percentiles, and the whiskers are the minimum to maximum values. P-values were determined by two-tailed Student's t-test. ***, $p < 0.001$; ns, not significant.

Reviewer #2:

In this study, the authors used DepMap data to build a co-dependency network to help identify novel factors that cooperate together for maintaining survival across diverse cell types. They then created their own analysis pipeline, Deplink, to link up these dependencies with cancer subtypes, drug dependencies, chromatin profiles, gene mutations and gene expression profiles. They validated their analyses using several approaches including highlighting the identification of known factors as well as using IP/westerns to validate a predicted interaction between C7orf26 and several integrator proteins.

They went on to concentrate on a module that contains KMT2A, MEN1 and several PRC2 complex components. Further analysis revealed that B cell lymphoma cell lines were particularly sensitive to loss of these factors, and function cooperatively. EZH2 gain of function mutant KARPAS422 cells had expanded H3K27me3 domains as well as reduced KMT2A and MEN1 binding profiles. They go on to show that EZH2 mutant cells are particularly sensitive to MEN1 inhibitors and that combined EZH2/MEN1 inhibition reduces tumour growth in a xenograft model.

This is a well written and clearly explained paper and I only have a few comments. There are a few details of the model and observed interactions that are not entirely clear.

We thank the reviewer for the positive comments.

1. What happens if you treat KARPAS422 cells with EZH2 inhibitors? Does this decrease H3K27me3 levels and expand KMT2A/MEN1 binding profiles? The model in Fig 6i might predict this.

We thank the reviewer for this excellent suggestion. We performed H3K27me3 CUT&Tag experiments with spike-in controls for *EZH2* mutant DLBCL line (Karpas422) upon EZH2 inhibitor (EPZ-6438) treatment at an early time point (72 hours). EZH2 inhibition effectively abolished genome-wide H3K27me3 enrichment, and regained MLL1/MEN1 bindings at previously H3K27me3-high regions (Supplemental Fig. 7a-b). Concomitantly, we observed decreased signal abundance of pre-existing MLL1/MEN1 peaks (Fig. 4e, Supplemental Fig. 7a-b). Together, these results suggest a causal function of H3K27me3 in regulating genome-wide patterns of MLL1/MEN1 binding.

Supplementary Fig. 7: (a) Heatmaps showing levels of MEN1/MLL1 binding and H3K27me3 around gained (top), shared (middle) and lost (bottom) MEN1 binding peaks upon EPZ-6438 treatment in KARPAS422 cells. Each row represents one MEN1 binding peak and shows CUT&Tag signal intensity (CPM) within 5 kb range on each side of the peak center.

(b) Integrative Genomics Viewer snapshot showing the landscape of H3K27me3, MEN1 and MLL1 binding at the CD24 (left), DDX5 (middle) and CDRT15P3 (right) gene loci in KARPAS422 cell treated with EPZ-6438 or DMSO.

Fig. 4e: Violin plots showing the normalized signal abundance (Z score) of MEN1 (left) and MLL1 (right) peaks at all MEN1/MLL1-bound promoters in Farage and KARPAS422 cells (all), or only those MEN1/MLL1-bound promoters shared between Farage and KARPAS422 cells (shared). EPZ, EZH2 inhibitor treatment using 1 μ M EPZ-6438 for 72 hr. The center line in the embedded boxplots represents the median, the box limits are the 25th and 75th percentiles, and the whiskers are the minimum to maximum values. P-values were determined by two-tailed Student's t-test. ***, $p < 0.001$.

2. What happens if you treat Farage cells with MEN1 inhibitors (the authors may want to consider the much more potent Syndax inhibitor that they use in Figure 6g rather than the weak MI-503), does this alter EZH2/H3K27me3 profiles? Or does the inhibition of binding only work in one direction?

We assessed H3K27me3 enrichment in Farage and KARPAS422 cells before and after the treatment of the 2nd generation MEN1 inhibitor VTP50469 using CUT&Tag. Indeed, we observed a gain of H3K27me3 surrounding MLL1/MEN1 binding peaks following MEN1 inhibitor treatment in both KARPAS422 and Farage cells (**Supplemental Fig. 7c-e**), suggesting that the opposition between MLL1-MEN1 complex and H3K27me3 is bidirectional. We thank the reviewer for suggesting this informative experiment.

Supplementary Fig. 7: (c) Heatmaps showing levels of MEN1/MLL1 binding and H3K27me3 around MEN1 binding peaks upon VTP-50469 treatment in Farage (left) and KARPAS422 cells (right). Each row represents one MEN1 binding peak and shows CUT&Tag signal intensity (CPM) within 5 kb range on each side of the peak center.

(d) Violin plots showing the normalized signal abundance (CPM) of MEN1 (left) and MLL1 (middle) peaks, and H3K27me3 (right) at all MEN1-bound promoters (within 5 kb range on each side of the promoter center) in Farage and KARPAS422 cells. VTP, MEN1 inhibitor VTP-50469. The center line in the embedded boxplots represents the median, the box limits are the 25th and 75th percentiles, and the whiskers are the minimum to maximum values. P-values were determined by two-tailed Student's t-test. ***, $p < 0.001$.

(e) Integrative Genomics Viewer snapshot showing the landscape of H3K27me3, MEN1 and MLL1 binding at the H4C11 (left), TDP2 (right) gene loci in Farage and KARPAS422 cells treated with VTP-50469 or DMSO.

3. Is there a reason why this is a B cell lymphoma specific effect? Does this dependency only exist where there is hyperactive EZH2? Does EZH2 inhibition in other cancer cell types (e.g. AML, ALL, others) cause a redistribution of KMT2A/MEN1 binding? Does KMT2A/MEN1 binding have reduced binding in all cell lines with higher levels of H3K27me3?

At the reviewer's comment, we further analyzed our Deplink result. We noticed that co-dependency module #8, which includes PRC1/2, MLL-MEN1 and KAT6/7 complexes, is preferentially essential for blood cancer cell lines harboring high levels of H3K27me3 (**Fig. 2c**). These include but are not limited to *EZH2* hyperactive mutant lymphomas. Interestingly, we did not observe a similar trend in solid tumor cell lines (**Fig. 2c**).

To experimentally verify this finding, we focused on multiple myelomas (MM). A subset of multiple myeloma (MM) tumors display globally elevated H3K27me3 levels due to either inactivation of H3K27 demethylases (UTX/KDM6A) or overexpression of PHF19. H3K27me3-high MM cell lines also show increased sensitivity to inhibition of EZH2. We treated a pair of MM cell lines that are either H3K27me3-high (RPMI-8226) or H3K27me3-low (MM.1S) (**Supplementary Fig. 4d**) with MEN1 inhibitor (VTP50469). MEN1 inhibition significantly reduced the viability of RPMI-8226 but not MM.1S cells in a dose-dependent manner (**Fig. 3g**), which confirms our Deplink analysis result.

Lastly, we determined if similar H3K27me3-driven redistribution of MLL-MEN1 complex binding can be observed in MM cells. We performed epigenomic profiling of H3K27me3-high RPMI-8226 and H3K27me3-low MM.1S cells. We found an increased number of H3K27me3 large domains in RPMI-8226 cells, which was accompanied by a decrease in the number of MEN1 but not MLL1 peaks (**Supplementary Fig. 6a-b**). Importantly, both genome-wide analysis and inspection at representative loci revealed higher signal abundance for MEN1/MLL1 promoter peaks in RPMI-8226 cells (**Supplementary Fig. 6c-d**), mirroring that in KARPAS422 cells. We also observed a similar "titration" of MLL1/MEN1 binding in RPMI-8226 cells following EZH2 inhibition (**Supplementary Fig. 6c**).

Based on these results, the reviewer is correct that the co-dependency between PRC2 and MLL-MEN1, and H3K27me3-driven redistribution of MLL-MEN1 complex binding, apply more broadly to hematopoietic tumors with high levels of H3K27me3 beyond B-cell lymphomas.

(Left) Supplementary Fig. 4d: Western blot showing H3K27me3 levels in MM cell lines and DLBCL cell lines, using β -actin as control. Relative abundance of H3K27me3 after control normalization of each cell line compared to that of MM.1S cell line is denoted.

(Right) Fig. 3g: Relative percentage of viable cells after 4-day treatment of MEN1 inhibitor VTP50469 with indicated dosage (n=3) for H3K27me3-high (RPMI-8226) and H3K27me3-low (MM.1S) multiple myeloma cell lines. Dots and whiskers are mean \pm s.d.

Supplementary Fig. 6: (a) Venn diagram showing the genomic partition (Mb) of H3K27me3 large domains (> 100 kb) shared between H3K27me3-high MM cell line RPMI-8226 (purple) and H3K27me3-low MM cell line MM.1S (green). (b) Venn diagrams showing the MEN1 (left) or MLL1 (right) binding promoters shared between RPMI-8226 (purple) and MM.1S (green) cells. (c) Violin plots showing the normalized signal abundance (Z score) of MEN1 (left) and MLL1 (right) peaks at all MEN1/MLL1-bound promoters in MM.1S and RPMI-8226 cells. EPZ, EZH2 inhibitor EPZ-6438. The center line in the embedded boxplots represents the median, the box limits are the 25th and 75th percentiles, and the whiskers are the minimum to maximum values. P-values were determined by two-tailed Student's t-test. ***, $p < 0.001$; ns, not significant. (d) Integrative Genomics Viewer snapshot showing the landscape of H3K27me3, MEN1 and MLL1 binding at the LRRC34 (left), LRRC26 (middle) and LAMTOR4 (right) gene loci in MM cell lines. EPZ, EZH2 inhibitor EPZ-6438.

Reviewer #3:

In this paper Lu et al. described a method for mapping the co-dependency using CRISPR-Cas9 KO fitness screens in order to quantify the gene-gene functional relationships. Modules identified in the co-essentiality networks are later associated with mutational, epigenome, gene expression and drug sensitivity of the cell lines in order to determine the mechanisms underlying essentiality of these modules. The results of this mapping and association can be used for discovering new trans-regulatory interactions as well as new targets for synthetic lethality. Using this resource authors show a surprising co-dependency between PCR2 and MLL-MEN1 and validated the functional relationship.

The paper describes an approach for identification of functionally similar genes (modules) using the co-essentiality networks. Although this is an interesting and promising idea, using the co-essentiality networks for identification of functions of uncharacterised genes has been used in many papers and recently a genome-wide study looked at the co-essential modules in this network for similar purposes. Here the main addition is the association of the modules with the genomic and epigenetic features of the cell lines (Deplink). Here are some small questions/comments related to the generation of the co-essentiality networks and their association with cell line features:

- The co-essentiality network is generated using 689 pan-cancer cell lines from Broad institute. There is a similar dataset from Sanger Institute as well as a combined data-set (published recently) where two datasets are normalised/bias removed and combined together. It would be important to generate the co-essentiality networks for the merged dataset. The combined data-set should give the most accurate representation of co-essentiality in these screens (better bias removal and more samples)

We have re-generated the co-dependency network using a combined CRISPR knockout screen data-set (covers 17,386 genes across 1,086 pan-cancer cell lines) derived from the combination of Broad's Achilles and Sanger's SCORE projects (DepMap release public 2022q2, normalized/bias removed). As a result, the updated analysis increased the number of our co-dependency modules from 84 to 145 (**Supplementary Fig. 1d**). We thank the reviewer for this suggestion that improved the accuracy of our analysis.

Supplementary Fig. 1d: Dependency correlation networks generated from the dependency correlation matrix using Genets (only top 15 strongest interactions for each node are shown). The 145 modules are numbered accordingly. The thickness of lines between genes indicates the correlation strength.

- In a related note it would be possible to compare the accuracy and robustness of these networks using the CORUM core complex member. It would be also nice to look at hu.MAP, a mass spectrometry (MS) protein–protein interaction database and STRING protein-protein interaction database for physical interactions only. The reliability of the co-essentiality network is quite important for the down-stream analysis.

We have mapped the biochemical complex interactions from CORUM database to the co-dependency network to validate the accuracy and robustness of our approach. Compared to randomly selected gene pairs in equal number ('simulated') as control, the dependency scores between CORUM-annotated complex members have significantly higher correlation ($p < 2.22e-16$) (Supplementary Fig. 1a).

We also overlapped the physical interactions collected in the BioGRID database to the genetic interactions in our network and 39% (305 out of 781) of our co-dependent genetic interactions can be identified in BioGRID datasets (Supplementary Table 1).

Supplementary Fig. 1a: Left, a schematic diagram showing the interactions mapped to reported protein biochemical interactions between CORUM core complex members as an observed dataset (purple) and randomly selected interactions in equal number as a simulated dataset (green). In total, 1,273 CORUM human core complexes are observed and simulated. Right, histogram of correlation scores of the observed and simulated complex-level interaction datasets. The basal positive correlation score (0.055) is indicated as a dashed line.

- Some of the essentiality modules are associated with some cancer types. It would be interesting to generate these co-essentiality networks separately for each cancer type and look at the associations within these networks. Demonstrating the same functional similarity in the related cancer types can be important and maybe similar associations can be identified in other cancer types.

We agree with the reviewer's comment and have attempted to generate cancer-type specific co-essentiality networks. However, for most cancer types, the analysis is statistically underpowered due to the limited number of cell lines. As an alternative approach, we performed unbiased clustering analysis for all cancer types as well as gene expression signatures based on dependency scores of our essentiality modules. Related cancer types (eg. blood cancers) and gene expression signatures (eg. Interferon alpha/gamma response) cluster more closely in this analysis (**Fig. 1b**).

Consistently, we found that the MLL-MEN1 and PRC2 co-dependency module (module #8) was significantly more essential to cell lines from DLBCL and closely related cancer types such as multiple myeloma (MM) (**Supplementary Fig. 4a**). We experimentally validated this finding and demonstrated that MEN1 inhibitor, alone or in combination with EZH2 inhibitor, was more toxic to H3K27me3-high MM cell line (RPMI-8226) when compared to H3K27me3-normal MM line (MM.1S) (**Fig. 3g**, **Supplementary Fig. 4d**, **Supplementary Fig. 9d**).

In addition, we have discussed the importance of future efforts to perform cancer type-specific co-essentiality mapping (Line 480-483).

Fig. 1: (b) Heatmap showing cancer type-specific (top) or cancer hallmark gene set enrichment analysis (GSEA) signature-specific (bottom) dependency of DCN modules. Names of cancer types are consistent with TCGA study abbreviations. For cancer type-specific dependency analysis, color scale shows the dependency score difference between cell lines from each specific cancer type and cell lines from other cancer types. Only modules significantly associated with at least one specific cancer type is shown (FDR < 0.1). For hallmark signature enrichment analysis, color scale shows the hallmark signature score difference between cell lines showing high dependency for each module and cell lines showing low dependency for that module (18% top and bottom cell lines ranked by dependency score, respectively, p-value < 0.01). Modules are ranked in the same order for both panels.

Supplementary Fig. 4a: Dot plot showing the dependency landscape of module #8 in cell lines across cancer types. Significant enrichments are denoted by asterisks (*, $p < 0.05$; ***, $p < 0.001$). Mean dependency scores are denoted by red bars.

(Left) Fig.3g: Relative percentage of viable cells after 4-day treatment of MEN1 inhibitor VTP50469 with indicated dosage (n=3) for H3K27me3-high (RPMI-8226) and H3K27me3-low (MM.1S) multiple myeloma cell lines. Dots and whiskers are mean \pm s.d.

(Middle) Supplementary Fig. 4d: Western blot showing H3K27me3 levels in MM cell lines and DLBCL cell lines, using β -actin as control. Relative abundance of H3K27me3 after control normalization of each cell line compared to that of MM.1S cell line is denoted.

(Right) Supplementary Fig. 9d: Relative percentage of viable MM.1S or RPMI-8226 cells following individual treatments of VTP50469 (160 nM, red), EPZ-6438 (200 nM, yellow), or combo treatment of both inhibitors (purple) for 4 days, normalized to the DMSO-treated controls (n=3). Bar plots and whiskers are mean \pm s.d. P-values were determined by two-tailed Student's t-test. *, $p < 0.05$

- While generating the co-essentiality networks only a subset of the genes (6464 genes) are selected. I think this is not justified quite well in the paper. I think using only a small portion of the genes for generating these networks is quite limiting for discovery purposes. Recently published co-essentiality papers all focus on the genome wide essentiality modules. Selecting a subset of the genes will affect the topology of the network quite a bit as well as the statistical analysis of the association of these modules with cell line features (number of modules will be different).

We have re-mapped our co-dependency network based on the reviewer's suggestion to include all 16,854 non-essential genes in this revision (revised workflow in Fig. 1a). While our major findings are not affected, as the reviewer predicted this indeed improved the network (eg. an increased number of co-dependent modules).

Fig. 1a: A workflow to set up the dependency correlation network (DCN) and Deplink analysis. Step 1, acquire the dependency profiles of 17,386 genes across 1,086 pan-cancer cell lines from DepMap CRISPR-Cas9 essentiality screen dataset; Step 2, for 16,854 non-essential genes, calculate the pairwise Pearson correlation score between each gene pair and generate the dependency correlation matrix; Step 3, generate DCN based on the correlation matrix using Genets; Step 4, integrate DCN with molecular profiles of pan-cancer cell lines using Deplink.

- Identification of the essentiality modules is pretty basic. Only genes that are co-essential with some correlation coefficient above certain threshold are used. Here also the subset of the genes play an important role. With a small number of genes, it is possible to get disconnected modules however in the genome wide analysis it will be harder to cluster the essentiality modules. However there are wide range of computational methods for detecting modules in networks (especially there are a lot of methods for co-expression networks) Analysing this data using the whole genome and some of these module detection methods would be more interesting and novel. This will also increase the scope and usefulness of the results as a resource for discovery.

In response to the reviewer's comments, we have revised the analysis to include all non-essential genes (see above). In addition, for the module detection from the co-essentiality network, we used a machine-learning algorithm Quack (integrated in GeNets) which was trained for networks based on cancer codependency relationships from project Achilles with good biological signals (AUC = 0.81) (Li et al., 2018, Nature Methods).

Integrating information from CORUM and BioGRID databases, we observed that direct physical interactions among complex subunits were often reflected by a strong correlation (Pearson's $r > 0.4$, such as C7orf26 and INTS10/13/14), and a moderate correlation (Pearson's r between 0.3 and 0.4) may reflect an indirect interaction for two proteins functioning in the same pathway (such as MLL-MEN1 and PRC2 complex members). To better visualize and prioritize gene-gene functional interactions, we used a two-step approach to develop our co-essentiality network by 1) first selecting genes ('core interactors') that have at least one strong interaction with other genes (Pearson's $r > 0.4$) and 2) then including genes that have moderate correlations with the 'core interactors' (Pearson's $r > 0.34$, which has optimized sensitivity and specificity on the recovery of CORUM complex-level interactions in **Supplementary Fig. 1b**). Nevertheless, we acknowledge that this cut-off strategy is still not perfect and arbitrary, and have addressed this limitation in the discussion (Line 451-455).

Supplementary Fig. 1b: ROC plot showing the sensitivity and specificity of identifying complex-level interactions using different cutoffs. The AUC scores and corresponding p-values (from the Student's t-test comparing the observed dataset under each cutoff and the simulated dataset) are indicated on the bottom right.

- The essentiality networks are associated with cell line features. Although mutational signatures and pathway expressions are used for associations, genetic variants such as SNPs, Indels and copy number are not used. It could be interesting to look at associations for specific variants or expression of individual genes.

We have associated essentiality modules with various features of pan-cancer cell lines, including gene expression, genetic mutation, chromatin modification, tumor mutation burden, microsatellite instability and copy number variation etc. Furthermore, we established a searchable database (<http://www.chaolulab-database.com/>) for readers to query detailed results. In the revised manuscript, we also highlight that a number of modules showed positive correlations with copy numbers and/or expression levels of module genes (**Supplementary Fig. 2e, f and Supplementary Table 13, 14**). We speculate that these modules represent cases of oncogene addition, where module genes are

amplified/overexpressed and required for tumor cell growth. One such example is module #53 (*SKP2/CDK2/CCNE1*), where there was a strong correlation between gene dependency scores and expression levels (**Fig. 1e**).

(Top left) Supplementary Fig. 2e: Volcano plot showing the correlation between gene-level copy number variation (CNV) data and dependency scores of gene members in each module across pan-cancer cell lines.

(Top Right) Supplementary Fig. 2f: Volcano plot showing the correlation between expression levels and dependency scores of gene members in each module across pan-cancer cell lines.

(Bottom) Fig.1e: Left, volcano plot showing the correlation between genetic dependency of DCN modules and drug sensitivity (GDSC). The cell lines with high dependency on module #53 (*CCNE1*, *CDK2*, *SKP2*) are more resistant to *CDK4/6* inhibitor Palbociclib. Right, the correlation between dependency and expression of genes in module #53.

- Deplink pipeline is not really well documented and seems to be pretty straightforward (for testing associations statistically). It will be nice to have more clarification about the novelty of this analysis.

The co-essentiality mapping has become a well-accepted approach for assigning uncharacterized genes to biological pathways. Our Deplink pipeline takes a step further and addresses why a module of co-dependent genes is specifically essential to a subset of cell lines but not others, thereby inferring the potential function of the module. For each module in the dependency correlation network, Deplink selects top and bottom cell lines based on their ranking of dependency scores of the module genes and compares their molecular features (chromatin modification, gene expression, genetic mutation, copy number variation, tumor mutation burden and microsatellite instability), various signatures (COSMIC, ISG, EMT, mRNAsi, GSEA hallmark), drug sensitivity and cancer types with those of the rest cell lines. To make the Deplink pipeline clearer, we made a tutorial page with detailed description of the code's functionality (<https://seanchen607.github.io/deplink.html>). We have added this information to the main text of revised manuscript.

- The results of the paper is pretty good and interesting. Identification of the new co-dependencies and their validation is quite interesting and important. The paper is written well and described the methods and results clearly.

We thank the reviewer for the positive comments.

Reviewers' Comments:

Reviewer #1:

Remarks to the Author:

The authors have done an extensive revision to address my previous concerns. I am satisfied with their new data and corresponding rebuttal. The current structure and logic flow improves much better. Although I am still a little bit concerning about the novelty, it is fair to let the audience to judge in the future. Other than that, I will congratulate the team for this great work.

Reviewer #2:

Remarks to the Author:

The authors have generated much new data and have fully answered my questions. The co-dependency of Men and PRC2 for cancer cell growth in specific subtypes is an important observation and will inform current and future drug discovery programs.

Reviewer #4:

Remarks to the Author:

Authors present Deplink, a computational pipeline to identify genetic co-dependency networks, based on high-throughput CRISPR knockouts screens, predict modules and associate disparate -omics data with these modules. Authors concentrate in one of the modules (module #8) and demonstrate the interplay between MLL-MEN1 and PRC2 complexes.

Authors have substantially improved the manuscript with new analyses that strength the results and predictions obtained with Deplink. However, while I have no comments about the experimental validation of the selected candidate, in line with Reviewers 1 and 3, I have concerns about the novelty of the computational pipeline. Which new insights are obtained with Deplink that are not obtained with existing approaches? The side-by-side comparison is limited to the work of Wainberg et al. 2022, which is a questionable decision, given the large number of approaches not included in the analysis, and that the differences are not statistically quantified. Moreover, the overlap between both approaches is large, particularly in Module #8, where they obtained 100% overlap. Thus, in my opinion, the question as to which new insights can be obtained with Deplink that cannot be extracted with existing methods has not been sufficiently addressed. Authors should concentrate on this point.

Minor comments

Why do authors use different Network cutoff for visualization purposes (0.4) and module identification (0.34)?

A brief description of Quack algorithm is recommended.

Point-by-point response to reviewer's comments:

Reviewer #1:

The authors have done an extensive revision to address my previous concerns. I am satisfied with their new data and corresponding rebuttal. The current structure and logic flow improves much better. Although I am still a little bit concerning about the novelty, it is fair to let the audience to judge in the future. Other than that, I will congratulate the team for this great work.

We thank the reviewer for the positive comments.

Reviewer #2:

The authors have generated much new data and have fully answered my questions. The co-dependency of Men and PRC2 for cancer cell growth in specific subtypes is an important observation and will inform current and future drug discovery programs.

We thank the reviewer for the positive comments.

Reviewer #4:

Authors present Deplink, a computational pipeline to identify genetic co-dependency networks, based on high-throughput CRISPR knockouts screens, predict modules and associate disparate -omics data with these modules. Authors concentrate in one of the modules (module #8) and demonstrate the interplay between MLL-MEN1 and PRC2 complexes.

Authors have substantially improved the manuscript with new analyses that strength the results and predictions obtained with Deplink. However, while I have no comments about the experimental validation of the selected candidate, in line with Reviewers 1 and 3, I have concerns about the novelty of the computational pipeline. Which new insights are obtained with Deplink that are not obtained with existing approaches? The side-by-side comparison is limited to the work of Wainberg et al. 2022, which is a questionable decision, given the large number of approaches not included in the analysis, and that the differences are not statistically quantified. Moreover, the overlap between both approaches is large, particularly in Module #8, where they obtained 100% overlap. Thus, in my opinion, the question as to which new insights can be obtained with Deplink that cannot be extracted with existing methods has not been sufficiently addressed. Authors should concentrate on this point.

We thank the reviewer for the suggestion to further clarify the novelty of our analysis pipeline. In **Supplementary Table S12** in the revised manuscript, we provide additional details of our methods and results, and compare them to that of similar recent studies of constructing pan-cancer genetic co-dependency networks from CRISPR-Cas9-based screening datasets.

	Pan, et al., 2018, Cell Systems PMID: 29778836	Boyle, et al., 2018, Molecular Systems Biology PMID: 30573688	Kim, et al., 2019, Life Science Alliance PMID: 30979825	Wainberg et al., 2021, Nature Genetics PMID: 33859415	The current work
CRISPR-Cas9-based screening datasets	Project Achilles	Project Achilles	Project Achilles	Project Achilles	Projects Achilles and SCORE
Number of pan-cancer cell lines	342	436	276	485	1,086
Module calling approach	Degree-preserved randomized network	Single cutoff for correlation coefficient (false discovery rate < 10%)	Single cutoff for correlation coefficient (Bonferroni-corrected $P < 0.05$ or Benjamini-Hochberg adjusted $P < 0.01$)	ClusterONE algorithm	Quack algorithm and two-step cutoff for correlation coefficient (matrix cutoff: Pearson's $r > 0.4$ and network cutoff Pearson's $ r > 0.34$)

Network visualization	igraph (R package)	igraph (R package)	Markov Cluster algorithm	Diffusion map	geNet (R package)
Computational pipeline for associating module dependencies and cancer molecular phenotypes	-	-	-	-	Deplink (R package)

Based on this comparison, we would like to highlight two major points:

(1) Comparing to pipelines from other studies, our current pipeline uses distinct approaches for module calling and network visualization. More importantly, our pipeline has better sensitivity and specificity (larger AUC in ROC curves) for predicting curated interactions in the BioGRID database (see below **Supplementary Fig. 10b**), which collects experimentally validated biochemical and genetic interactions. We have also revised text to address this point in the discussion.

The reviewer is correct that Module #8 is 100% covered by Wainberg et al., 2021. However, 18 modules (among 5229 modules) in their prediction contain both MEN1 and PRC2 complex members (each module contains 64 members on average). Therefore, we believe our modules have higher confidence and specificity based on the more stringent cut-off.

(2) A key novelty of our analysis (i.e. **Deplink**) is to associate co-dependency modules with specific phenotypic and molecular features of cancer cell lines. To our knowledge, only Wainberg et al. has correlated co-dependency modules to cancer tissue types. Our Deplink analysis, in contrast, is much more comprehensive and results in the identification of co-dependency modules that are specific to cancer mutational signatures, gene expression signatures, chromatin modifications and drug sensitivity. As such, our analysis reveals new insights not reported elsewhere, including the predicted synthetic lethal interactions of *PRPF39-TRNAU1AP* gene pair in mismatch repair-deficient cancer cells, and INO80 complex in histone acetylation-high cancers, just to name a few.

Lastly and respectfully, we would like to point out that the construction of co-dependency network represents only a small part of our results (Fig. 1A). The majority of the manuscript (Figures 2-6) report original findings

about the functional validation and mechanistic characterization of the co-dependency between MLL-MEN1 and PRC2 complexes in lymphomas.

Minor comments

1. Why do authors use different Network cutoff for visualization purposes (0.4) and module identification (0.34)?

As proteins usually function in a complex and one complex could further functionally interact with other complexes, we came up with a complex-centric strategy using two-step cutoff for correlation coefficient to enhance the power of discovering novel functional modules. We first use a more stringent cutoff (Pearson's $r > 0.4$) to identify complex-level interactions, and then use a less stringent cutoff (Pearson's $|r| > 0.34$) as the network cutoff to help capturing/visualizing the less strong genetic/functional interactions between members of distinct complexes (see below **Supplementary Fig. 10a**). We have also revised text to include this point in the discussion.

For example, in module #8, the average correlation coefficients between members within MLL-MEN1 and PRC2 complexes are 0.560 and 0.674, respectively, while the average correlation coefficient between two complexes is 0.391. The matrix cutoff (Pearson's $r > 0.4$) used in the first step will predict members from MLL-MEN1 and PRC2 complexes into two complex-level communities. The network cutoff (Pearson's $|r| > 0.34$) in the second step recognizes these two biochemically distinct complexes as one functionally interacting module.

2. A brief description of Quack algorithm is recommended.

In response to the reviewer's suggestion, we added a brief description of Quack algorithm in the section of Materials and Methods:

Based on the correlation matrix, the dependency correlation network was generated and visualized using GeNets and a R package 'geNet', which integrates a machine-learning algorithm Quack that is trained for comparing the global and local biological signal of networks and identifying the optimal network with which to interpret large genomic datasets such as cancer co-dependency relationships from project Achilles (Li et al., 2018, Nature Methods).

Reviewers' Comments:

Reviewer #4:

Remarks to the Author:

Authors have convincingly addressed all my questions. Congratulations!

Point-by-point response to reviewer's comments:

Reviewer #4:

Authors have convincingly addressed all my questions. Congratulations!

We thank the reviewer for the positive comments.